# Visualizing the Diversity of Representations Learned by Bayesian Neural Networks

**Dennis Grinwald**                                                          *dennis.grinwald@tu-berlin.de*
*Machine Learning Group*
Technical University of Berlin, Berlin, Germany
BIFOLD – Berlin Institute for the Foundations of Learning and Data, Berlin, Germany

**Kirill Bykov**                                                               *kbykov@atb-potsdam.de*
*Understandable Machine Intelligence Lab*
Leibniz Institute for Agriculture and Bioeconomy (ATB), Potsdam, Germany
Technical University of Berlin, Berlin, Germany
BIFOLD – Berlin Institute for the Foundations of Learning and Data, Berlin, Germany

**Shinichi Nakajima**                                                          *nakajima@tu-berlin.de*
*Machine Learning Group*
Technical University of Berlin, Berlin, Germany
BIFOLD – Berlin Institute for the Foundations of Learning and Data, Berlin, Germany
RIKEN Center for AIP, Tokyo, Japan

**Marina M.-C. Höhne**                                                         *mhoehne@atb-potsdam.de*
*Understandable Machine Intelligence Lab*
Leibniz Institute for Agriculture and Bioeconomy (ATB), Potsdam, Germany
Technical University of Berlin, Berlin, Germany
BIFOLD – Berlin Institute for the Foundations of Learning and Data, Berlin, Germany

**Reviewed on OpenReview:** `https://openreview.net/forum?id=ZSxvyWrX6k`

## Abstract

Explainable Artificial Intelligence (XAI) aims to make learning machines less opaque, and offers researchers and practitioners various tools to reveal the decision-making strategies of neural networks. In this work, we investigate how XAI methods can be used for exploring and visualizing the diversity of feature representations learned by Bayesian Neural Networks (BNNs). Our goal is to provide a global understanding of BNNs by making their decision-making strategies a) visible and tangible through feature visualizations and b) quantitatively measurable with a distance measure learned by contrastive learning. Our work provides new insights into the *posterior* distribution in terms of human-understandable feature information with regard to the underlying decision-making strategies. The main findings of our work are the following: 1) global XAI methods can be applied to explain the diversity of decision-making strategies of BNN instances, 2) Monte Carlo dropout with commonly used Dropout rates exhibit increased diversity in feature representations compared to the multimodal posterior approximation of MultiSWAG, 3) the diversity of learned feature representations highly correlates with the uncertainty estimate for the output and 4) the inter-mode diversity of the multimodal posterior decreases as the network width increases, while the intra-mode diversity increases. These findings are consistent with the recent Deep Neural Networks theory, providing additional intuitions about what the theory implies in terms of humanly understandable concepts.

# 1  Introduction

Despite the great success of Deep Neural Networks (DNN), little is known about the decision-making strategies that they have learned. This lack of transparency is a major cause of concern since DNNs are being used in safety-critical applications (Berkenkamp et al., 2017; Huang & Chen, 2020; Le et al., 2021) and it has been shown that they tend to encode fallacies, including the memorization of spurious correlations (Lapuschkin et al., 2019; Izmailov et al., 2022) or being biased towards the data set that they were trained on (Tommasi et al., 2017; Turner Lee, 2018; Ribeiro et al., 2016). Therefore, in recent years, the field of Explainable Artificial Intelligence (XAI) has established itself in order to make the decisions of AI models comprehensible to humans. XAI methods allow the user to "open" the black-box of trained neural networks, that is, understanding what the model has learned and on which features its decisions are based.

Such *post-hoc* XAI methods can be further categorized into *local* and *global* explanation methods. While *local* XAI methods assign importance scores to input features, e.g. image pixels, that are important for a model's prediction (Sundararajan et al., 2017; Simonyan et al., 2014; Lapuschkin et al., 2015; Rs et al., 2017; Smilkov et al., 2017), *global* XAI methods aim to explain the inner workings of a DNN, e.g. what concepts are encoded in its parameters, by providing *feature visualizations* (FVs) (Olah et al., 2017; 2018; Bau et al., 2020). XAI methods have been applied extensively on many Deep Neural Network types that were trained in a *frequentist* fashion, namely Convolutional Neural Networks (Lecun et al., 1998), Recurrent Neural Networks (Medsker & Jain, 1999), and others (Hilton et al., 2020; Schnake et al., 2021; Bau et al., 2020).

In a recent work by Bykov et al. (2021), explanation methods have been applied to neural networks that were trained in a Bayesian fashion. Inherently, Bayesian Neural Networks (BNN) offer the property of uncertainty estimation that results from the diversity of learned feature representations and prediction strategies of different BNN instances. The authors apply *local* explanation methods on top of BNN instances and obtain a distribution over explanations, from which they estimate explanation uncertainties. Furthermore, such an approach of introducing stochasticity to the model parameters was shown to be successful in enhancing local explanations of standard models trained in a *frequentist* fashion (Bykov et al., 2022).

In our work, our primary objective is to explore the feasibility of utilizing *global* explanation methods for Bayesian Neural Networks. We aim to answer the following questions:

1. Can the diversity of BNN instances be explained by *global* XAI methods?

2. Does the choice of the Bayesian inference method affect the diversity of their feature visualizations?

3. Can the uncertainty estimates provided by a BNN be explained by the diversity of their feature visualizations?

4. How does the network width affect the diversity of explanations of samples from a multimodal posterior distribution?

The first question is to make sure that the properties of Bayesian ensembles can be visualized. The other three questions are related to hot topics in Bayesian Deep Learning, i.e., scalable posterior approximation, reliable uncertainty estimation, and the loss surface of Deep neural networks that have been exploited in deep ensembles (Lakshminarayanan et al., 2017) and their extensions (Wilson & Izmailov, 2020). By answering those questions, we show the capability of our approach in analyzing Bayesian ensembles of neural networks. To the best of our knowledge, we are the first to use *global* XAI methods to explain the diversity of BNN instances. Our results underpin the latest findings in the field of deep learning theory (Roberts et al., 2022) in a – for the first time – illustrative way.

# 2  Background

In this section, we provide a brief overview of Bayesian Neural Networks and global explanation methods, which are used in the subsequent sections.

## 2.1 Bayesian neural networks

Bayesian machine learning estimates the posterior distribution over unknown variables rather than estimating a single value. We focus on neural network classifiers, where the conditional distribution of the model is given as

$$p(y|x, \theta) = \text{SoftMax}(f_\theta(x)), \tag{1}$$

with $f_\theta(x)$ being the network output parameterized with $\theta$. Let $\mathcal{D} := \{(x^{(1)}, y^{(1)}), \ldots, (x^{(N)}, y^{(N)}) | x^{(n)} \in \mathbb{R}^D, y^{(n)} \in \{0, 1\}^C \ \forall n = 1, \ldots, N\}$ be a training data set, where $N$ is the number of training samples, and $C$ is the number of distinct classes. The label $y^{(n)}$ is a one-hot encoded vector, where the vector-entry corresponding to the correct class is one and the rest is set to zero. Given a *prior* distribution $p(\theta)$ on the network parameter, the *posterior* distribution is given by Bayes' theorem (Pearson et al., 1959):

$$p(\theta|\mathcal{D}) = \frac{p(\mathcal{D}|\theta)p(\theta)}{p(\mathcal{D})} = \frac{p(\mathcal{D}|\theta)p(\theta)}{\int p(\mathcal{D}|\theta)p(\theta)d\theta}. \tag{2}$$

Then, the *predictive* distribution $p(y^*|x^*, \mathcal{D})$ for a test point $x^*$ is given by marginalizing over all possible model parameter setups:

$$p(y^*|x^*, \mathcal{D}) = \int p(y^*|x^*, \theta)p(\theta|\mathcal{D})d\theta, \tag{3}$$

encoding the models uncertainty regarding the predicted label $y^*$ for a given test point $x^*$.

Since the integrals in Eqs.(2) and (3) are computationally intractable for DNNs, several approximation methods have been proposed (Mackay et al., 1999; Neal, 2012; Graves, 2011; Blundell et al., 2015; Zhang et al., 2019). Those methods approximate the posterior (2) by $q(\theta|\mathcal{D}) \approx p(\theta|\mathcal{D})$, and estimate the predictive (3) with $T$ Monte Carlo samples:

$$p(y^*|x^*, \mathcal{D}) \approx \frac{1}{T} \sum_{t=1}^{T} p(y^*|x^*, \theta_t), \quad \theta_t \sim q(\theta|\mathcal{D}). \tag{4}$$

The resulting averaged output probability vector of the BNN is usually referred to as Bayesian Model Average (BMA).

Below we introduce four popular methods for approximating the posterior with unimodal or multimodal distribution families.

### 2.1.1 Kronecker-Factored Approximation Curvature (KFAC)

KFAC (Martens & Grosse, 2015) approximates the true posterior distribution by learning the following *unimodal* multivariate normal distribution:

$$q_{\text{MAP}}(\theta|\mathcal{D}) = \mathcal{N}(\theta; \hat{\theta}_{\text{MAP}}, \hat{F}^{-1}),$$

where $\mathcal{N}(\mu, \Sigma)$ denotes the Gaussian distribution with mean $\mu$ and covariance $\Sigma$. The approximate posterior mean $\hat{\theta}_{\text{MAP}}$ is the *Maximum A Posteriori* (MAP) estimate, which is obtained by a standard training algorithm such as Stochastic Gradient Descent (SGD). The posterior covariance $\hat{F}^{-1}$ is the inverse of a regularized approximation of the Fisher information matrix $F$. For computational efficiency, the inverse Fisher information matrix $\hat{F}^{-1}$ is approximated by the sum of a diagonal matrix and a low-rank matrix expressed as a Kronecker factorization.

### 2.1.2 Monte-Carlo Dropout (MCDO)

Deep Neural Networks of arbitrary depth including non-linearities that are trained with dropout can be used to perform approximate Bayesian inference (Gal & Ghahramani, 2016). In MCDO, samples from the approximate posterior are drawn by

$$\theta_l = \theta_l^* \cdot diag([z_{l,m_l}]_{m_l=1}^{M_{l-1}}), \quad z_{l,m_l} \sim \text{Bernoulli}(\gamma_l),$$

where $\theta_l^*$ is the weight matrix of layer $l$ of dimensions $M_l \times M_{l-1}$ before Dropout is applied, $diag(\cdot)$ is a function that maps the vector $[z_{l,m_l}]_{m_l=1}^{M_{l-1}}$ to a diagonal matrix of dimension $M_{l-1} \times M_{l-1}$ with $z_{l,m_l}$ being the diagonal element in row $m_l$ and column $m_l$ for $m_l = 1, ..., M_l$. The elements $z_{l,m_l}$ are independent binary random variables that follow the Bernoulli distribution, e.g. $z_{l,m_l} \sim \text{Bernoulli}(\gamma_l)$ and $\gamma_l$ is the dropout rate. In practice, users can choose a fixed value of $\gamma_l$ that optimizes some metric, e.g. the test accuracy.

### 2.1.3 Stochastic Weight Averaging Gaussian (SWAG)

SWAG (Maddox et al., 2019) is an efficient method to fit a Gaussian distribution around a local solution. The algorithm consists of an initial training phase followed by a model collection phase. The initial training is performed in a standard way, e.g., by SGD with a decaying learning rate. After the convergence, the collection phase starts, where the SGD update continues with either a cyclical or a high constant learning rate for collecting SGD iterates. The trajectory of those iterates is then used to estimate the mean and the covariance of the approximate Gaussian posterior distribution:

$$q_{\text{SWAG}}(\theta|\mathcal{D}) = \mathcal{N}(\theta; \hat{\theta}_{\text{SWAG}}, \hat{\Sigma}_{\text{SWAG}}).$$

The covariance $\hat{\Sigma}_{\text{SWAG}}$ is approximated by the sum of a diagonal and a low-rank matrix, similarly to KFAC, for stable estimation from a small number of SGD iterates during the collection phase.

### 2.1.4 Deep Ensemble of SWAG (MultiSWAG)

MultiSWAG (Wilson & Izmailov, 2020) combines the ideas of SWAG and deep ensemble (Lakshminarayanan et al., 2017). It simply performs the SWAG training multiple times from different weight initializations, and each SGD trajectory during the collection phase is used to compute the mean $\hat{\theta}_k$ and the covariance $\hat{\Sigma}_k$ of a Gaussian distribution. The estimated Gaussians are combined to express the approximate posterior as a (equally-weighted) mixture of Gaussians (MoG):

$$q_{\text{MultiSWAG}}(\theta|\mathcal{D}) = \frac{1}{K}\sum_{k=1}^{K}\mathcal{N}(\theta; \hat{\theta}_k, \hat{\Sigma}_k). \tag{5}$$

Since each SWAG trajectory is expected to converge to a different local solution or mode, MultiSWAG provides a *multimodal* approximation of the true *posterior*. MultiSWAG was shown to improve generalization performance and uncertainty estimation quality (Wilson & Izmailov, 2020).

## 2.2 Global XAI methods

XAI methods that are decoupled from the architectural choice of a neural network and its training procedure are referred to as *post-hoc* XAI methods, which can be further categorized into *local* and *global* explanation methods. *Global* XAI methods aim to explain the general decision-making strategies learned by the representations of DNNs. They reveal the concepts to which a particular neuron responds the most (Olah et al., 2017; 2018; Bau et al., 2020; Nguyen et al., 2016) by decomposing and quantifying the activations of certain neural network layers in terms of human-understandable concepts (Kim et al., 2018; Koh et al., 2020), or by identifying and understanding causal relationships that are encoded between neurons (Reimers et al., 2020).

In this work, we focus on the Activation Maximization (AM) framework (Nguyen et al., 2016). The general idea of AM is to artificially generate an input that maximizes the activation of a particular neuron in a certain layer of a neural network. The optimization problem can be formulated as follows:

$$\hat{v} = \underset{v \in \mathbb{R}^V}{\text{argmax}}\; a(v) + R(v), \tag{6}$$

where $v$ is the input variable, $a(\cdot)$ is the activation of the neuron of interest, and $R(\cdot)$ is a regularizer. This can be easily extended to maximize the activation of a certain channel or layer by maximizing a norm of the channel's or layer's activation vectors. We refer to the resulting image, $\hat{v}$ in Eq.(6), as a Feature Visualization (FV) vector. To generate FV that are not full of high-frequency noise, as is the case for

| Data set | image size | classes | Training size | Test size |
|----------|-----------|---------|--------------|-----------|
| Places365 | $256 \times 256 \times 3$ | 365 | ~1.4mil | 365k |
| CIFAR-100 | $32 \times 32 \times 3$ | 100 | 50k | 10k |
| STL-10 | $96 \times 96 \times 3$ | 10 | 5k | 8k |
| SVHN | $32 \times 32 \times 3$ | 10 | ~73k | ~26k |

Table 1: Data sets used in the experiments.

| Model | # Trainable parameters | Test accuracy |
|-------|----------------------|---------------|
| ResNet50 | 25,557,032 | 55.57% |
| WRes10 | 36,546,980 | 82.75% |
| WRes2 | 1,481,252 | 77.45% |
| WRes1 (WideResNet28) | 376,356 | 72.57% |
| WRes0.7 | 88,448 | 62.54% |
| WRes0.2 | 5,008 | 27.96% |

Table 2: Network architectures. WRes1 corresponds to the original WideResNet28, and WRes$\beta$ for $\beta \neq 1$ is the network with the number of channels $\beta$ times more than the original in each convolutional layer. The test accuracy is on Places365 for ResNet50, and on CIFAR-100 for WRes$\beta$, respectively.

adversarial examples (Szegedy et al., 2013; Goodfellow et al., 2014; Athalye et al., 2018), several regularization techniques have been proposed (Olah et al., 2017; Mordvintsev et al., 2018): *transformation robustness* applies several stochastic image transformations, e.g., jittering, rotating, and scaling, before each optimization step; *frequency penalization* either explicitly penalizes the variance between neighboring pixels or applies bilateral filters on top of the input. In order to even further reduce high-frequency patterns that correspond to noise, it was proposed to perform the optimization in a spatially decorrelated and whitened space, instead of the original image space. This space corresponds to the Fourier transformation of the image based on the spatially decorrelated colors. In this way, high-frequency components are successfully reduced.

Feature Visualization methods have proven effective in explaining the concepts learned within Deep neural Networks (Goh et al., 2021). These methods have also been employed for the purpose of explaining the *circuits* within DNNs, which represent computational subgraphs responsible for the transformation of various features (Cammarata et al., 2020). Additionally, Activation Maximisation approaches have been used to identify neurons responsible for *spurious correlations* (Bykov et al., 2023) and human-implanted backdoors (Casper et al., 2023).

# 3 Experimental Setup

Here we describe our experimental setup including the methodologies for the quantitative analysis of feature visualizations.

## 3.1 Datasets

In our experiments, we use four data sets, Places365 (Zhou et al. (2017)), CIFAR-100 (Krizhevsky et al. (2009)), STL-10 (Coates et al. (2011b)), and SVHN (Yuval (2011)), of which some statistics are listed in Table 1. When training a classifier, the Places365 images are clipped to $224 \times 224 \times 3$, and the other data sets are clipped to $32 \times 32 \times 3$. Places365 dataset encompasses numerous classes with diverse human-understandable concepts, such as the farm class, which typically includes concepts like farm animals, barns, fields, and others. In computationally intensive experiments with MultiSWAG, we use CIFAR-100 and clipped images of STL-10 and SVHN.

| Inference method | Test accuracy |
|---|---|
| KFAC | 54.78% |
| MultiSWAG | 56.64% |
| MCDO-5% | 55.68% |
| MCDO-10% | 55.07% |
| MCDO-25% | 51.69% |

Table 3: Approximate Bayesian inference methods analyzed in the experiments. For each inference method, the test accuracy of ResNet50 on the Places365 test set is evaluated on the predictive distribution that we approximate with $T = 50$ MC samples.

## 3.2 Network Architectures

We train ResNet50 (He et al., 2016) on Places365, and WideResNet28 (Zagoruyko & Komodakis, 2016) on CIFAR-100, respectively. To study the network width dependence, we also use WideResNet28 with increased and decreased numbers of channels in each layer. Therefore, we scale the number of channels of the original WideResNet28 by a scaling factor $\beta$, which we refer to as WRes$\beta$. For example, WRes2 corresponds to a network that is twice as wide as the original network. The number of parameters and the test accuracy are shown in Table2, where the test accuracies are obtained by evaluating the MAP estimates of the corresponding models.

## 3.3 Inference Methods

The approximate Bayesian inference methods that we analyzed are listed in Table 3. MCDO-$\gamma$% is the MC dropout model with dropout rate $\gamma$%, applied to each layer of the CNN encoders except for the last one (same dropout rate for all layers). For MultiSWAG, the approximate posterior is a mixture (5) of $K = 10$ Gaussians. In Table 3, the test accuracy of ResNet50 on the Places365 test set with each inference method is evaluated according to Eq.(4) with $T = 50$ MC samples.

## 3.4 Feature Visualization

In all experiments, we use the AM framework (6) to obtain the FV vectors and analyze their behavior. The FV vector is by default of the same size as the input image. However, since the images in CIFAR-100 are small and therefore not very informative in terms of the concepts that we can extract from them qualitatively, we expand the input to $128 \times 128 \times 3$ using bilinear interpolation in PyTorch. We solve the AM optimization problem (6) by 512 steps of gradient descent with the step size of $\alpha = 0.05$. For regularization, we apply the transformation robustness with random rotation, random scaling, and random jittering. Moreover, the optimization is performed in the decorrelated and whitened space. All transformations correspond to the default setting in Olah et al. (2017), for which we used the PyTorch (Paszke et al., 2019) version of the published source code available at `https://github.com/greentfrapp/lucent`.

## 3.5 Quantitative Distance Measure in FV space

One of the main contributions of this work is to analyze the diversity of the "concepts", expressed in FV vectors, of BNN instances *quantitatively*. To this end, we need to define a distance measure in the space of FVs. Apparently, the standard norm distances, e.g. L2-distance and cosine-similarity, directly applied to the FV vectors are not appropriate since "concepts" in FV should be invariant to translations and rotations. We, therefore, use a non-linear function $g : \mathbb{R}^V \mapsto \mathbb{R}^Z$ that maps FV vectors into a low-dimensional *latent concept space* such that FVs with similar concepts are mapped to close points. Afterward, the distance between two FVs is measured by the cosine-similarity in this space:

$$d(v, v') = \frac{g(v)^\top g(v')}{\|g(v)\|\|g(v')\|}. \tag{7}$$

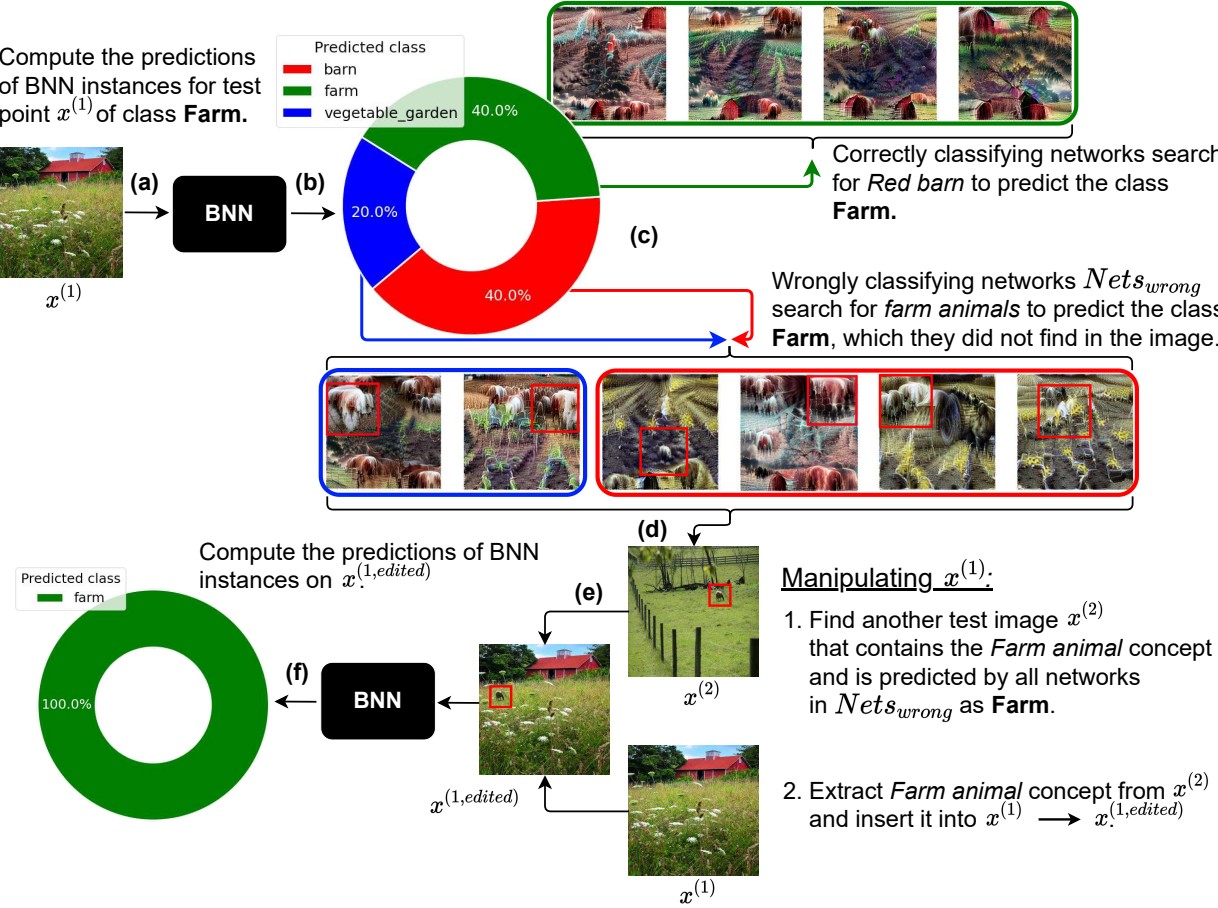

Figure 1: Explaining decision-making strategies of 10 individual BNN instances using FVs. (a) A test sample $x^{(1)}$ of class *Farm*. (b) 10 BNN instances classify $x^{(1)}$. (c) Classification results by 10 instances (only 40% classify $x^{(1)}$ correctly as *Farm*). FVs of the "correct" instances $N_{\text{correct}}$ are shown in the upper green box, while those of the "wrong" instances $N_{\text{wrong}}$ are in the lower blue and red boxes. We observe that FVs of Nets$_{\text{wrong}}$ contain *Farm animal*-like concepts, which they did not find in the input $x^{(1)}$. (d) Another test sample $x^{(2)}$ that contains a tiny sheep. (e) We cut out the tiny sheep (*Farm animal* concept) patch from $x^{(2)}$ and paste it into $x^{(1)}$, yielding $x^{(1,\text{edited})}$. (f) All instances correctly classify $x^{(1,\text{edited})}$ as *Farm*, which implies that the FVs indeed encode human-understandable concepts that reflect the network's decision-making strategy.

We learn the function $g$ via a contrastive learning scheme (Chen et al., 2020; Li et al., 2020; 2021), and more specifically, we use the SimCLR framework (Chen et al., 2020).

Assume that we are given a training set $\{v_n\}_{n=1}^N$ of FVs. We first conduct stochastic data augmentation, which applies $M$ random image transformations to each FV, resulting in $M$ different versions to be used as an augmented training set $\{\{\widetilde{v}_{n,m}\}_{m=1}^M\}_{n=1}^N$. Here $\{\widetilde{v}_{n,m}\}_{m=1}^M$ denotes the generated samples by stochastic augmentation applied to the $n$-th original sample, i.e., $v_n$. Then, the non-linear map $g_\phi(\cdot)$ parameterized by $\phi$ is trained by minimizing the contrastive loss:

$$\sum_{n=1}^N \sum_{m,m'=1}^M \log \frac{\exp\left(\frac{d_\phi(\widetilde{v}_{n,m},\widetilde{v}_{n,m'})}{\tau}\right)}{\sum_{n'\neq n}\sum_{m'',m'''=1}^M \exp\left(\frac{d_\phi(\widetilde{v}_{n,m''},\widetilde{v}_{n',m'''})}{\tau}\right)}, \tag{8}$$

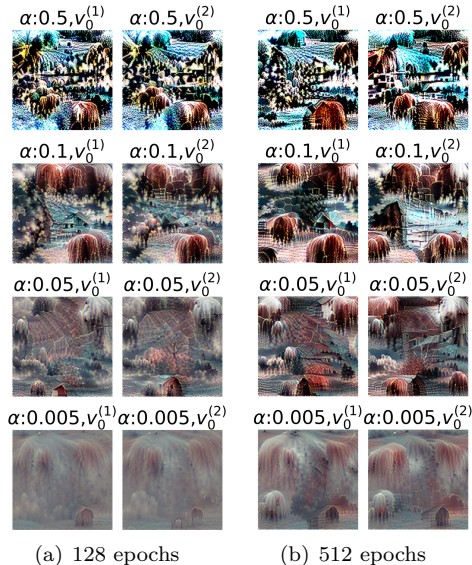

(a) 128 epochs          (b) 512 epochs

Figure 2: Dependence of the FV on the number of epochs, the learning rate $\alpha$, and the parameter initialization ($v_0^{(1)}$ or $v_0^{(2)}$). We observe that the number of epochs affects the color scheme slightly, that the learningn rate $\alpha$ affects the crispness and contrast drastically, and that the parameter initialization affects the positions and shapes of objects slightly, while the high-level concepts stay the same for different initializations.

where $d_\phi$ is the distance measure (7) which depends on $\phi$ through the mapping $g_\phi(\cdot)$, and $\tau$ is a temperature hyperparameter. This contrastive loss aims to minimize the distances between the samples transformed from the same original test sample while maximizing the distances between those transformed from different original test samples. For the architecture of $g_\phi$, we use a ResNet18 (He et al., 2016) base encoder, that is, all ResNet18 layers up to the average pooling layer, followed by a fully-connected layer, a *ReLU* non-linearity and another fully-connected layer. We train one contrastive learning model per Bayesian inference method. In particular, we generate 1000 FVs per BNN, except for MultiSWAG. In MultiSWAG, we generate 100 FVs per ensemble member and train our distance on the concatenation of their FVs, thus resulting in 1000 total FVs. The number of augmented samples is $M = 2$ (per original sample and epoch), and the augmentation is randomly chosen from "Random Cropping", "Colorjitter", and "Horizontal Flipping". We use stochastic gradient descent (SGD), where the contrastive loss (8) with the batch size $N$ and the temperature $\tau = 0.5$ is minimized in each epoch. We run SGD for 150 epochs on the CIFAR-100, STL-10, and SVHN models, and for 250 epochs on the Places365 models. For the contrastive learning, we used the implementation of the following public repository available at `https://github.com/Yunfan-Li/Contrastive-Clustering`.

## 4   Experimental Results

In this section, we visualize and analyze the diversity of the decision-making strategies of BNN instances in terms of high-level concepts by using feature visualizations. The first three experiments are dedicated to validating our approach. We first demonstrate that the extracted FVs properly reflect the characteristics of each individual BNN model instance by relating FV to the classification prediction, and observing the model's uncertainty behavior when manipulating test samples. Then, we analyze the dependence between FVs and the initial parameter setting of the AM algorithm. Finally, we analyze the correlation between the diversity of FVs and the uncertainty estimates of BNN model instances. While the first two experiments validate our methodology qualitatively, the third experiment confirms quantitatively that FVs reasonably reflect the property of a Bayesian ensemble.

In the fourth and the fifth experiments, we use our distance measure to analyze the diversity of BNN instances from different posterior approximation methods, and for different scales of the model, which provide us with new insights into the latest findings of deep learning theory (Roberts et al., 2022). To the best of

our knowledge, no previous work has used *global* XAI methods to explain the behavior of Bayesian Neural Network instances.

## 4.1 Visualizing characteristics of individual BNN instances

The prediction by a BNN is made from an ensemble of model instances drawn from the (approximate) posterior distribution, and such model instances can acquire different decision-making strategies. In this first experiment, we qualitatively show that by using FVs we are able to explain the diverse characteristics of each individual model instance in the Bayesian posterior ensemble. We draw model instances from the MCDO-5% posterior with the ResNet50 architecture trained on Places365. To receive a global explanation for each of the 10 BNN instances for the class *Farm*, we apply the AM algorithm to the network output, maximizing the logit for the label *Farm*. From the 10 different FV images, shown in the green, blue and red boxes in Figure 1, we can observe that all FV images contain reasonable high-level concepts, e.g., *Red barn*, *Farm animal*, *Tractor*, *Crop field*, and *Pasture fence*, implying that all BNN instances learned reasonable decision-making strategies for the class *Farm*. However, we identify some test samples which are misclassified by some of the BNN instances.

One of the wrongly classified test samples, $x^{(1)}$, is shown in Figure 1(a). Indeed, 4 of the instances classify $x^{(1)}$ correctly as *Farm*, while the other 6 instances classify it wrongly as either *Barn* or *Vegetable_garden* as shown in the pie chart in Figure 1(c). The corresponding FVs of the BNN instance are plotted next to the pie chart, arranged based on their prediction: the 4 FVs in the upper green box correspond to the instances classifying the input correctly (which we refer to as Nets$_{correct}$), while the 6 FVs in the lower blue and red boxes correspond to those classifying the input wrongly (which we refer to as Nets$_{wrong}$). The networks that correspond to the FVs in the blue box wrongly predict the input as *Vegetable garden*, while the ones that correspond to the FVs in the red box wrongly predict the input as *Barn*. Comparing the FVs of Nets$_{correct}$ and Nets$_{wrong}$, we notice that most of the FVs of Nets$_{wrong}$ contain fur or farm animal-like objects such as sheep, horses and cows, while FVs of Nets$_{correct}$ do not. This implies that the 6 instances in Nets$_{wrong}$ use animals within their decision strategy to classify an image as *Farm*, and since $x^{(1)}$ does not contain any *Farm animal* concept, they can not classify it correctly. To further demonstrate that FVs can reveal the characteristics of each BNN instance, we manipulate the test sample $x^{(1)}$ by using another test sample $x^{(2)}$ which contains a very small sheep (*Farm animal* concept) and which is classified correctly by all BNN instances in Nets$_{wrong}$. From $x^{(2)}$, we cut out the small sheep and paste it into $x^{(1)}$ manually (see $x^{(1,\text{edited})}$ in the figure). Now, all instances classify $x^{(1,\text{edited})}$ correctly as *Farm* as shown by the green pie chart. This implies that the high-level animal concepts inherent in the FVs of Nets$_{wrong}$ are indeed the concepts that the networks are searching for in the input image in order to classify an input as *Farm*.

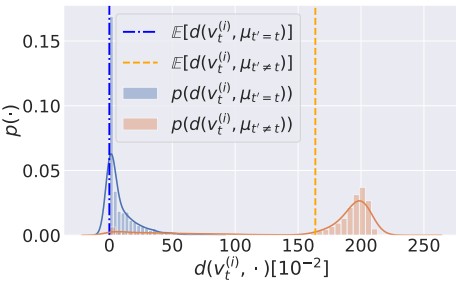

Figure 3: Distribution of distances between FVs and their corresponding instance centers (blue) and distribution of the distances between different instance centers (orange). The distances within the instances are much smaller than between different instances, and both distributions are clearly separated. The impact of parameter initialization for the AM optimization is ignorable.

Two other similar examples are given in Appendix A. With this experiment we demonstrated, that the diverse decision-making strategies of different BNN model instances can indeed be explained by global XAI methods, providing evidence for our question 1) from the introduction.

## 4.2 Dependence on hyperparameters and initialization

The AM optimization (6) is highly non-convex, and therefore the obtained FVs can depend on the hyperparameter setting and parameter initialization. Hence, we would need to appropriately choose the setting such that the FVs properly extract the concepts that the networks are using for the decision-making process. Accordingly, we investigate the dependence of FVs on the hyperparameters, e.g., the number of epochs for SGD and the step size $\alpha$, and the initialization — we start from two random initial points, $v_0^{(1)}$ and $v_0^{(2)}$, to solve the optimization problem (6). From the results, shown in Figure 2, we can observe that the AM optimization for $\alpha = 0.005$ does not converge even after 512 epochs, and $\alpha = 0.5$ results in over-contrasted

FVs. We furthermore observe that different initializations can change the location and shape of the concept objects slightly, while the content of the represented concepts is unchanged for different settings and initializations. This result implies that our approach can be used for analyzing the decision-making process without carefully tuning the parameters. Note that the location shift of concept objects by initializations might strongly affect the quantitative analysis of the diversity of BNN instances if we would adopt a naive distance measure, e.g., the L2 or cosine distance in the pixel space. We will make sure in the next experiment that this is not the case when we use the distance measure introduced in Section 3.5.

### 4.3 Quantitative diversity of FVs

Here, we validate our quantitative distance measure in the FV space by showing that it suffices two requirements: 1) the diversity caused by parameter initializations for the AM optimization is ignorable compared to the diversity of BNN instances, and 2) the measured diversity is highly correlated to the uncertainty of the prediction. In order to evaluate 1), we first generate 5 FVs for each of 100 BNN instances and compute the average *latent concept vector* for each of the instances. We will refer to these 100 mean vectors as instance centers. In Figure 3 we plot the histogram of the Euclidean distance from each FV to its corresponding instance center (blue), as well as to another instance center (orange). We can observe a significant separation between the two histograms, implying that the FV diversity caused by initialization is indeed ignorable.

Next, we investigate the correlation between the diversity of FVs and the predictive entropy,

$$H(x^*) := -\sum_{c=1}^{C} P(y^* = c | x^*, \mathcal{D}) \log P(y^* = c | x^*, \mathcal{D}),$$

which is a measure for uncertainty. We prepare 100 sets, each comprised of 100 model instances with different FVVar (see Appendix B for how to generate those sets), and plot the FV variance

$$\text{FVVar} := \frac{1}{T} \sum_{t=1}^{T} \left\| g_\phi(v_t) - \frac{1}{T} \sum_{t'=1}^{T} g_\phi(v_{t'}) \right\|_2^2 \tag{9}$$

in the horizontal axis and the empirical mean predictive entropy over the whole data set $\mathcal{D}$

$$\mathbb{E}_{p(x)}[H(x)] := \frac{1}{N} \sum_{n=1}^{N} H(x^{(n)})$$

in the vertical axis in Figure 4. Here, $T$ is the number of BNN instances in a set, $\{v_t\}_{t=1}^{T}$ are the FVs of them,

and the predictions (for computing the entropy) are made by those $T$ instances. We can observe a high correlation (0.83) between the FVs diversity and the uncertainty estimates, showing that our distance measure in the FV space properly reflects the distance between BNN instances. Overall, we can conclude that our distance measure satisfies the two requirements above, and we now apply our tools for analyzing BNN instances.

### 4.4 Comparing representations of different BNN inference methods

Here, we will visually and quantitatively compare the learned representations of models that were trained using different Bayesian inference methods. We train ResNet50 on Places365 with the Bayesian approximation methods listed in Table 3. First, we find the

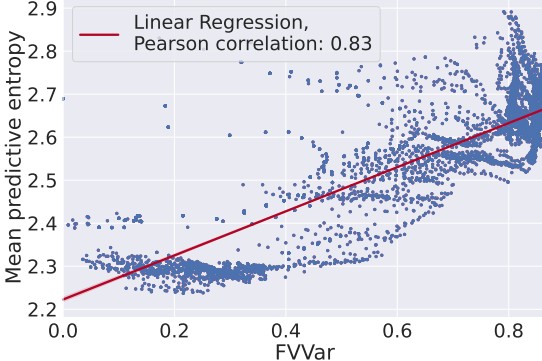

Figure 4: Linear regression and Pearson correlation between feature visualization diversity (FVVar) and mean predictive entropy. High correlation between FVVar and the mean predictive entropy is observed.

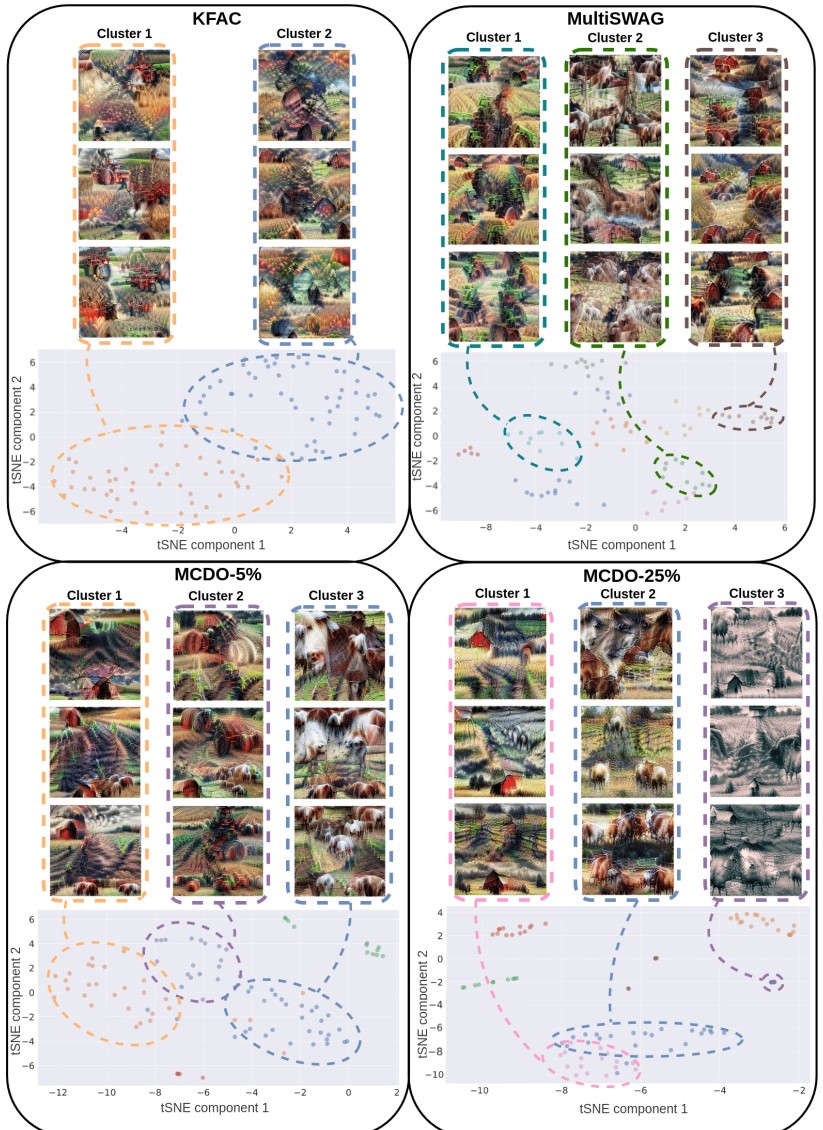

Figure 5: FVs and cluster structure of the learned concepts for class logit *Farm*. Four black boxes represent KFAC, MultiSWAG, MCDO-5%, and MCDO-25%, respectively. The FVs are clustered using KMeans (where the number of clusters is manually chosen) and plotted in two-dimensional tSNE space in different colors. We choose certain clusters and plot 3 example FVs of each cluster. The FVs in a colored rectangle are from the cluster depicted as an ellipse in the same color in the tSNE plot. For each Bayesian inference method, we mostly found the following concepts in the displayed clusters. **KFAC** cluster 1: *Tractor*, *Crop field*, and little details of animals, e.g. *Eyes*. KFAC cluster 2: *Red barn*, *Crop field*, *Pasture fence*, and little details of animals, e.g. *Eyes*. **MultiSWAG** Cluster 1: *Tractor*, *Crop field*. MultiSWAG cluster 2: *Farm animal*, *Crop field*, and *Pasture fence*. MultiSWAG cluster 3: *Red barn* and *Crop field*. **MCDO-5%** cluster 1: *Red barn*, *Crop field*. MCDO-5% cluster 2: *Tractor* and *Crop field*. MCDO-5% cluster 3: *Farm animal*, *Pasture fence*. **MCDO-25%** cluster 1: *Red barn*, *Crop field*, *Pasture fence*. MCDO-25% cluster 2: *Farm animal*, *Pasture fence*, and *Crop field*. MCDO-25% cluster 3 mostly contains a mixture of all found *Farm* concepts in a specific grayscale.

cluster structure of BNN instances of each inference method and compare their learned concepts. To this end, we compute the FV for the class *Farm* of each of the 100 BNN instances, individually. Figure 5 shows tSNE plots of the BNN instances in the FV space with typical FV images from each cluster for KFAC,

MultiSWAG, MCDO-5%, and MCDO-25%. Note that tSNE is applied in the latent concept space, i.e., to the vectors $\{g_\phi(v_t)\}$, and therefore reflect our quantitative distance measure. Clustering is performed by applying KMeans (Lloyd, 1982) (again to the latent concept vectors $\{g_\phi(v_t)\}$), and the instances that belong to different clusters are depicted in different colors. We can observe that the concepts of the class *Farm*, which are visually perceptible, generally consist of *Red barn*, *Tractor*, *Farm animal*, *Crop field*, and *Pasture fence*. However, different inference methods yield different cluster structures: KFAC yields 2 clusters and MultiSWAG yields $\sim 10$ clusters indicated by the different colors in Figure 5 respectively. Furthermore, we can observe that MCDO yields $5 \sim 7$ clusters, while the distribution looks highly dependent on the dropout rate. KFAC yields the least diverse FVs, in terms of visual comparison, with 2 clusters, and all instances tend to include almost all of the *Farm* concepts. Nevertheless, we observe a difference between the 2 equally sized clusters which relates to the *Tractor* concept being more present in cluster 1 and the *Red barn* concept being more present in cluster 2.

The MultiSWAG instances also include different *Farm* concepts in each of the instances. However, visualizing their clusters individually, we can observe that some clusters include certain concepts more frequently than others. In particular, cluster 1 includes the *Tractor* and *Crop field*, cluster 2 the *Farm animal* and *Pasture fence*, and cluster 3 the *Red barn*, *Crop field* and *Pasture fence* concepts more frequently than the other clusters. The other 7 clusters contain all *Farm* concepts and are similar in terms of the content of concepts (see Appendix C). For MultiSWAG, the clusters found by KMeans match the MoG structure, i.e., most of the BNN instances generated from the same posterior Gaussian component are clustered together. This connects to the fact that each SWAG ensemble member converges to a different local minimum (Wilson & Izmailov, 2020), or posterior mode. MCDO-$\gamma$ shows the most diverse FVs. We observe in Figure 5 that the BNN instances of each cluster seem to specialize with respect to certain *Farm* concepts and can be thus separated very well by these concepts. For MCDO-5%, cluster 1 primarily includes the *Red barn* and *Crop field*, cluster 2 the *Tractor* and *Crop field*, and cluster 3 the *Farm animal* and *Pasture fence* concepts. Naturally, the diversity of MCDO-$\gamma$ instances increases with increasing Dropout rate, and thus MCDO-25% results in an increased number of clusters. Also, we observe that the quality of FVs decreases with increasing Dropout rate and that more diverse color schemes appear, e.g. two yellow scales, two gray scales, and one blue scale for MCDO-25%. In Appendix C, we additionally show the results for MCDO-10% and include examples of the remaining clusters of the MultiSWAG model instances. The right plot in Figure 6 shows the quantitative diversity of FVs, i.e., FVVar defined in Eq. (9), which behaves consistently with our qualitative observations: KFAC yields the lowest variance, followed by MultiSWAG, and the MCDO 5%, 10%, 25% yields the largest diversity in this order. Hence, we can answer question 2) from the introduction "*Does the choice of the Bayesian inference method affect the diversity of their feature visualization?*" with yes.

The right subplot in Figure 6 compares the FV diversity and the mean predictive entropy, which exhibits a clear correlation. We observe, for the first time, that the FV diversity correlates with the predicted uncertainty estimates. This answers question 3) from the introduction "*Can the uncertainty estimates provided by a BNN be explained by the diversity of their feature visualizations?*".

## 4.5 Visualizing the multimodal structure of the posterior distribution of BNNs.

Here we explain the multimodal structure of the BNN posterior distribution. Specifically, we use BNN instances drawn from the MoG posterior (5) obtained by MultiSWAG, and qualitatively (visually) and quantitatively analyze their behaviors. Furthermore, we investigate the dependence of the multimodality on the network width in terms of humanly understandable concepts using FVs. To this end, we train a WideResNet28 and its modified versions listed in Table 2, where WRes$\beta$ refers to a WideResNet28 network with the width scaled by $\beta$. The models are trained on CIFAR-100 by MultiSWAG with a mixture of $K = 10$ Gaussians posteriors. After training, we draw 100 BNN instances from each Gaussian posterior, resulting in 1000 BNN instances in total, and compute their individual FVs for the class *Castle*. Figure 7 shows distributions of the 1000 BNN instances in the FV space, where the rows correspond to the networks with different widths, i.e., WRes0.2, WRes1, and WRes10, respectively (Results with other network widths are shown in Appendix D). For each row, a tSNE plot of FVs is shown in the bottom, and FVs of three BNNs from three hand-picked modes are visualized in the top. Note that the color in the tSNE plot indicates the Gaussian component of MultiSWAG (KMeans is not applied here). From the tSNE plots, we observe that

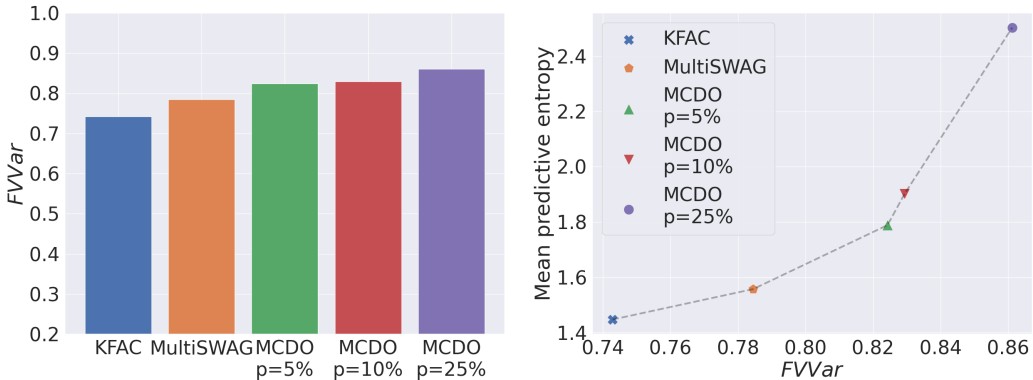

Figure 6: Representation diversity of Bayesian inference methods. The left subplot displays the FVVar of BNN instances of each inference method. The diversity of KFAC, MultiSWAG, MCDO 5%, 10%, and 25% is increasing from left to right. The right subplot displays the correlation between FVVar and the mean predictive entropy. They highly correlate with each other, as expected.

the network width strongly affects the multimodal structure: For small network width (WRes0.2), instances from different modes are separable, while, for wide network (WRes10), the instances are overlapped.

For quantifying this observation, we compute the inter-mode variance

$$\text{InterModeVar} := \frac{1}{K} \sum_{k=1}^{K} \left\| \mu_k - \frac{1}{K} \sum_{k'=1}^{K} \mu_{k'} \right\|_2^2 \tag{10}$$

and the intra-mode variance

$$\text{IntraModeVar} := \frac{1}{K} \sum_{k=1}^{K} \frac{1}{100} \sum_{t=1}^{100} \left\| z_{k,t} - \mu_k \right\|_2^2, \tag{11}$$

and plot them in the left panel of Figure 8. Here, $z_{k,t}$ is the latent concept vector of the FV of the $t$-th instance from the $k$-th mode ($k$-th Gaussian component), and $\mu_k$ is the latent concept vector of the FV of the $k$-th Gaussian center. We can observe that, with a growing network width, the inter-mode variance decreases, while the intra-mode variance increases. The former (decreasing variance) aligns with the implications of recent theory on Neural Tangent Kernels (NTK) (Jacot et al., 2018; Arora et al., 2019), while the latter (increasing intra-mode variance with growing $\beta$) has not been explained by theory, to the best of our knowledge. In order to show that the observed tendency is not because of the limited expressivity of the ResNet18 that we used to define our distance measure, nor the dataset-specific properties, we conducted the same experiments with ResNet34 and ResNet50 as the backbone networks and on other datasets including STL-10 (Coates et al., 2011a) and SVHN (Netzer et al., 2011), and observed a similar tendency in Appendix E. We also confirm in Appendix F that whether starting from pre-trained models or training from scratch does not significantly affect the properties of posterior distribution.

In the following, we will answer question 4) from the introduction regarding the impact of the network width on the diversity of FV of samples from a multimodal posterior distribution. To this end, we first qualitatively investigate the FVs given in Figure 7. We can observe that a too narrow network (WRes0.2) gives notably low-quality FVs, which implies that the network does not have sufficient capacity to learn good feature representations. The large inter-mode variance reflects the fact that each mode learns different color schemes and different patterns, while the small intra-mode variance results in identical concepts within each mode. For larger network widths, we observe a clear difference in their FVs. While the modes of WRes1 still learn different color schemes, and at the same time also learn high-level *Castle* concepts, e.g. *castle towers*, the modes of WRes10 learn very similar features, and it is harder to distinguish between the FVs of different modes. The difference between WRes1 and WRes10 implies that, for a moderate network width, each mode plays different roles while for a large network width, modes get mixed and each mode abstracts the concepts well, such that a single model alone can perform classification well.

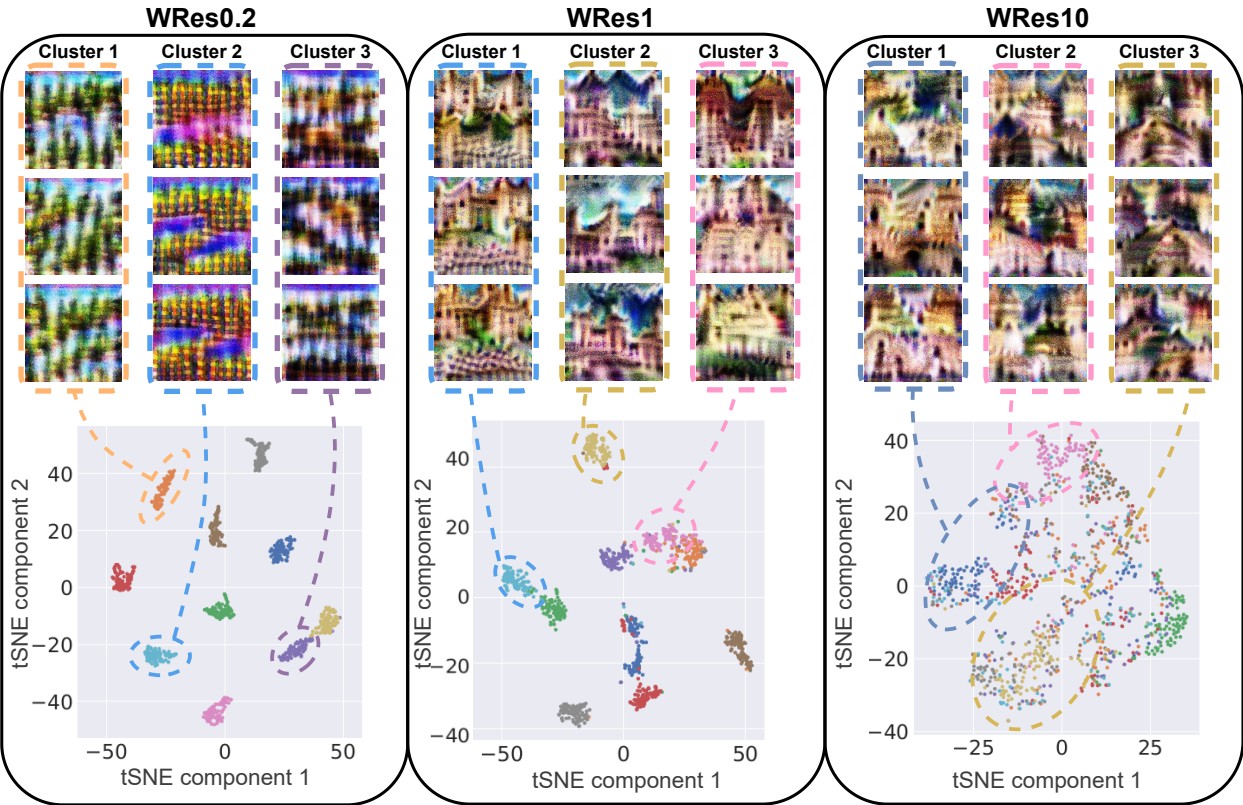

Figure 7: Multimodal structure of BNNs trained by MultiSWAG in the FV space. The rows correspond to the networks with different widths, i.e., WRes0.2, WRes1, and WRes10, respectively. For each network, a tSNE plot of 1000 BNN instances is shown in the bottom, and FVs of three BNNs from three hand-picked SWAG modes are visualized in the top. We observe that the modes tend to overlap as the network width increases. Moreover, the quality of FVs drastically improves from WRes0.2 to WRes1, and WRes10 successfully abstracts the castle class by replacing discrete shape information (small discrete shapes, e.g., in the bottom of WRes1 Cluster 1) with smoother ones in each mode (compare the bottom of WRes10 Cluster 2).

We confirm this implication by quantitatively evaluate how strongly *ensembling*, e.g., with deep ensemble and MultiSWAG, improves the performance. In the right panel of Figure 8, we plot the performance gain by MultiSWAG compared to its single-mode counterpart, i.e., SWAG, as a function of the network width. More specifically, the red curve shows the test accuracy of MultiSWAG (i.e., the test accuracy of the averaged model) subtracted by the average accuracy over the separate SWAG models (i.e., the average of the test accuracies of the models). We observe that the performance gain by ensembling is largest for WRes1, and decreases when the network width further increases.

The cyan curve similarly shows the performance gain in terms of the test expected calibration error (ECE) (Guo et al., 2017). Namely, the curve shows ECE of MultiSWAG subtracted by the average of the ECEs over the separate SWAG models. Noting that the lower ECE is the better, we see that ensembling degrades the uncertainty estimation performance when the network width is small, e.g., WRes0.2 and WRes0.7, while it significantly improves the test ECE when the network width is large, e.g., WRes2 and WRes10. We will investigate this phenomenon further in our future work.

Our extensive experiments revealed how the width of the underlying network architecture affects the FVs, answering question 4) in the introduction "*How does the network width affect the diversity of explanations of samples from a multimodal posterior distribution?*".

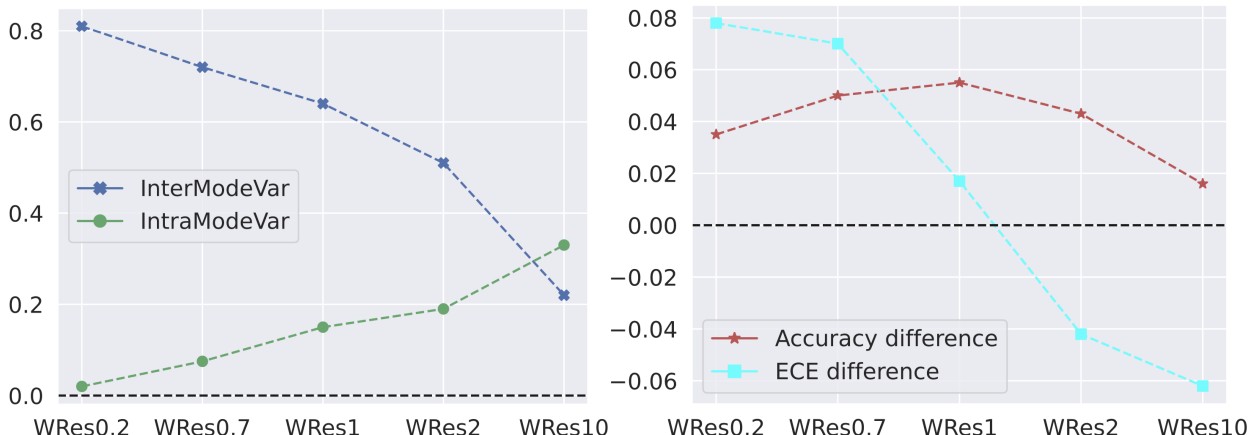

Figure 8: The left subplot shows how the inter- and intra-mode variances change with increasing network width. As the network width increases (from left to right), the inter-mode variance decreases, while the intra-mode variance increases. The right subplot shows the performance gain by *ensembling* in terms of the test accuracy, as well as the test expected calibration error (ECE), i.e., the test accuracy (ECE) of MultiSWAG subtracted by the average accuracies (ECEs) over the separate SWAG models. Noting that the higher accuracy (lower ECE) is the better, we observe that for larger network widths (WRes2 and WRes10), the performance gain decreases for the test accuracy, while it increases for the uncertainty estimation.

## 5 Conclusion

Since BNNs provide additional information about the uncertainty of a prediction, they are of enormous value, especially in safety-critical applications. Their ability to estimate uncertainties of a prediction is inherent in the learned multimodal posterior distribution. Sampling from this posterior distribution results in BNN instances, exhibiting diverse representations, which in turn lead to different prediction strategies. It has been shown in a large number of works that the diversity of these strategies depends on various factors, such as the choice of the Bayesian approximation method, the parameter initialization, or the model size. However, so far, this diversity has been analyzed either in the output, or parameter space of the BNN instances, which unfortunately still lacks human understandable intuition. With this work, we now deliver this missing but important building block to support human understanding by making the learned strategies visually accessible. To this end, we use feature visualizations as a global explanation method to explain — in a human-understandable way — the different representations and prediction strategies learned by BNN instances. Furthermore, this enables us to examine the diversity of the BNN instances on the feature visualizations both qualitatively and quantitatively, thus adding the visual component to the previous analysis.

In order to quantitatively analyze the FVs with their pronounced heterogeneity, we first learn a suitable representation of the FV with the help of contrastive learning, which we can then use to measure the distance. The ability to measure the distances between FVs allows us to investigate and at the same time to visually understand how the use of different Bayesian inference methods affects the diversity of BNN instances. Indeed, we could demonstrate, that the learned representations vary stronger for multimodal Bayesian inference methods, such as MultiSWAG than for unimodal ones, such as KFAC. The greatest variety of learned representations is achieved by Dropout-based models. Here the dropout rate correlated positively with the variety of representations, i.e. the higher the dropout rate is, the more different the representations visible through their FVs are. Furthermore, we showed that the diversity of FVs of BNN samples is positively correlated with the uncertainty estimates that we obtain from this BNN.

We were also able to measure — and visually demonstrate — the dependence of the multimodal structure of the posterior distribution on the width of the underlying network. Specifically, we have shown that the modes in a multimodal posterior distribution of MultiSWAG become more similar with increasing width

of the underlying network. This result is consistent with recent theoretical insights into Neural Tangent Kernels, where it was shown that the local solutions of infinitely wide networks behave similarly. By adding the additional visual component - through the lens of *global* explanations - we can easily understand the similar behavior of the modes given their similar FVs. In future work, we will investigate how the observed behavior of *posterior* modes can help to improve model performance in detecting OOD samples.

## Acknowledgements

This work was partly funded by the German Ministry for Education and Research (BMBF) through the project Explaining 4.0 (ref. 01IS200551), the German Research Foundation (ref. DFG KI-FOR 5363), the Investitionsbank Berlin through BerDiBa (grant no. 10174498), and the European Union's Horizon 2020 programme through iToBoS (grant no. 965221), and BIFOLD - Berlin Institute for the Foundations of Learning and Data (ref. BIFOLD23B).

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

## A  Decision-making strategy manipulation

In the introductory experiment in Section 4.1, we demonstrate that we can visualize and extract high-level information about the decision-making strategies of BNN instances from FV and, for the first time, visualize their differences. Here we show a few other examples in Figure 9.

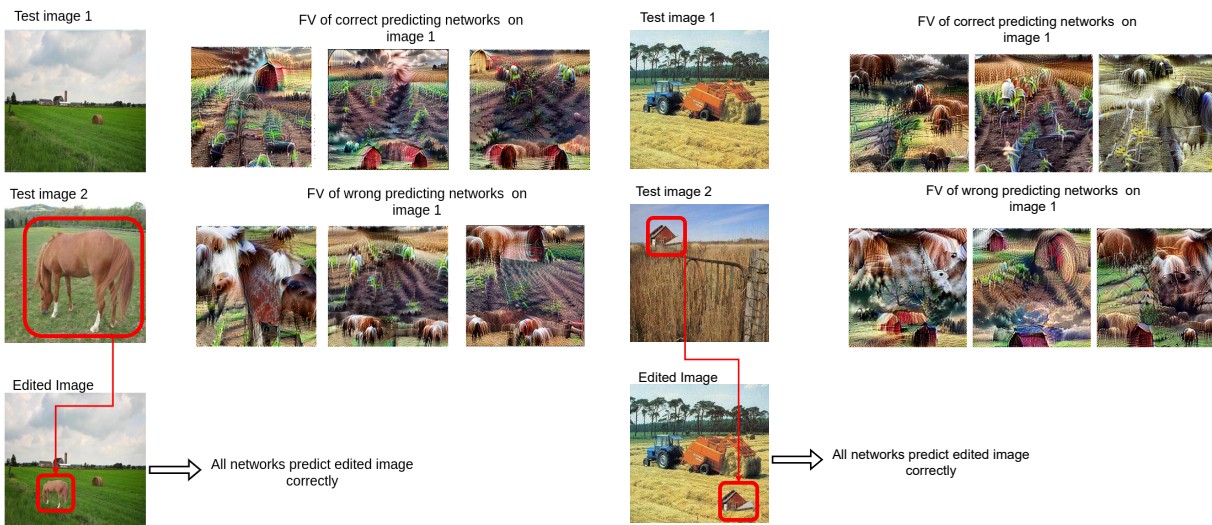

Figure 9: Two additional image manipulation examples. **Left:** In this example we can see that the selected images that predicted "Test image 1" correctly include the high-level concepts *Red barn* and *Crop fields*, which are also present in the image. On the other hand, the chosen networks that predicted the image incorrectly, include the high-level concept *Farm animal* in them. After manipulating Test image 1 by pasting a horse (*Farm animal*) from "Test image 2" of the test set, all networks do predict the "Edited image" correctly. **Right:** In this example we can see that the selected images that predicted "Test image 1" correctly include the high-level concept *Crop field*, which is also present in the image. On the other hand, the chosen networks that predicted the image incorrectly, include the high-level concept *Red barn* in them. After manipulating Test image 1 by pasting a red barn from "Test image 2" of the test set, all networks do predict the "Edited image" correctly.

## B  Constructing FV diversity sets

Here, we explain how we formed the sets of BNN instances used in Section 4.3, where the correlation between the FV diversity, FVVar, and the mean predictive entropy is evaluated. We first prepared a pool $\Theta := \{\theta_t | t = 1, ..., T_{total}\}$ of BNN instances, by drawing samples from the posterior distribution. From $\Theta$, we generated 100 different sets $\{S_i | i = 1, \ldots, 100\}$, each of which consists of 100 BNN instances. Each set $S_i$ collects samples from $\Theta$ in the following way: after randomly choosing the first instance $\theta_{S_i}^1 \in \Theta$, we iteratively add the nearest neighbor (in the FV metric space) of the last added instance $\theta_{S_i}^{t-1}$ for $t = 1, \ldots, 100$. Note that, every time we add an instance to a set, the corresponding instance is removed from the pool, i.e., $\Theta \leftarrow \Theta \setminus \theta_{S_i}^t$. Although we did not control the diversity of each set, the resulting sets had different diversity as shown in Figure 4.

## C  Clustering representations of different BNN inference methods

For the experiments in Section 4.4 we cluster the FVs by applying KMeans. We choose the number of clusters by qualitatively analyzing the goodness of clusters, that is, whether the points cluster well in the tSNE plots and whether human-understandable concepts, e.g. *Farm animal*, are clustered together. We show the tSNE plots for the MCDO-10% model in Figure 10, the FVs of MCDO-10% in Figure 11, and the FVs of the MultiSWAG clusters in Figure 12.

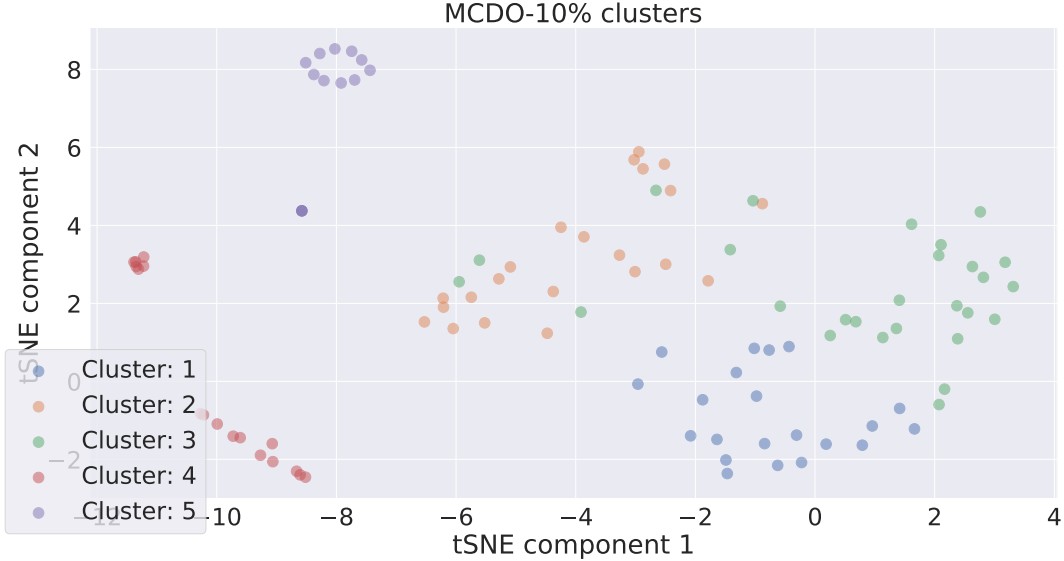

Figure 10: MCDO-10% clusters. The instances cluster into 5 clusters, of which two, Clusters 1 and 4, are well separated, while the other three, Clusters 2, 3, and 5, are connected.

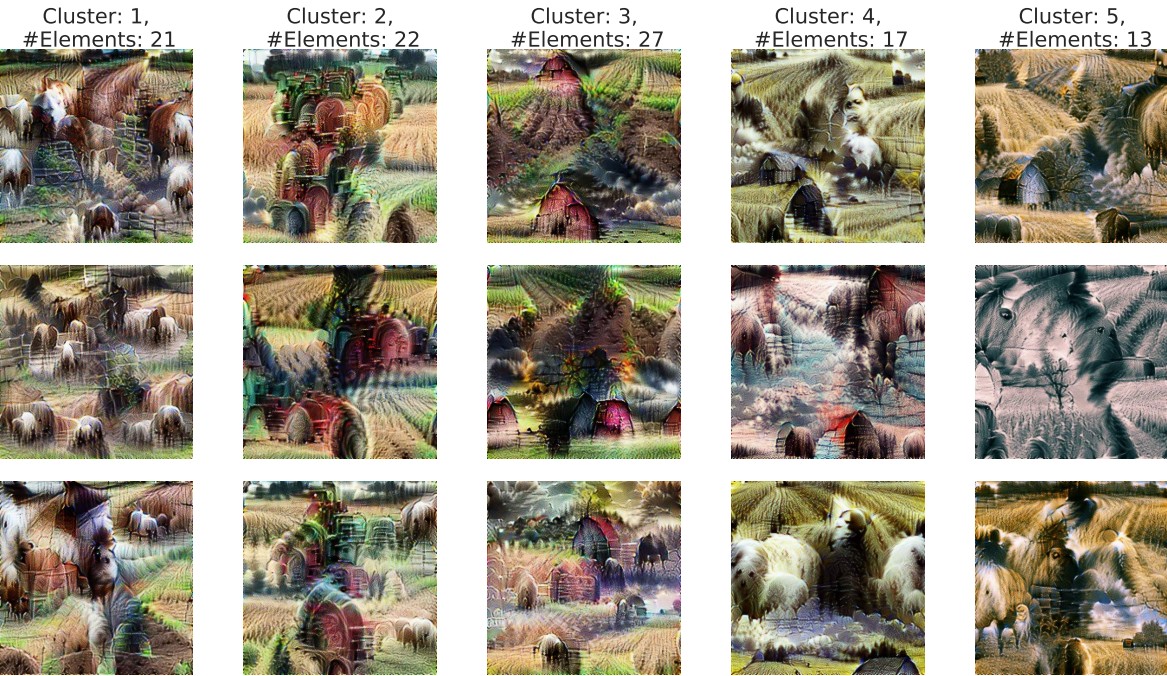

Figure 11: Five clusters formed by MCDO-10%. Clusters 1-3 are connected, and similar in terms of their color scheme. However, Cluster 1 contains the *Farm animal*, Cluster 2 the *Tractor*, and Cluster 3 the *Red barn* concepts more frequently than the others. The other two clusters, Clusters 4 and 5, contain a mix of all *Farm* concepts, however in a different color scheme.

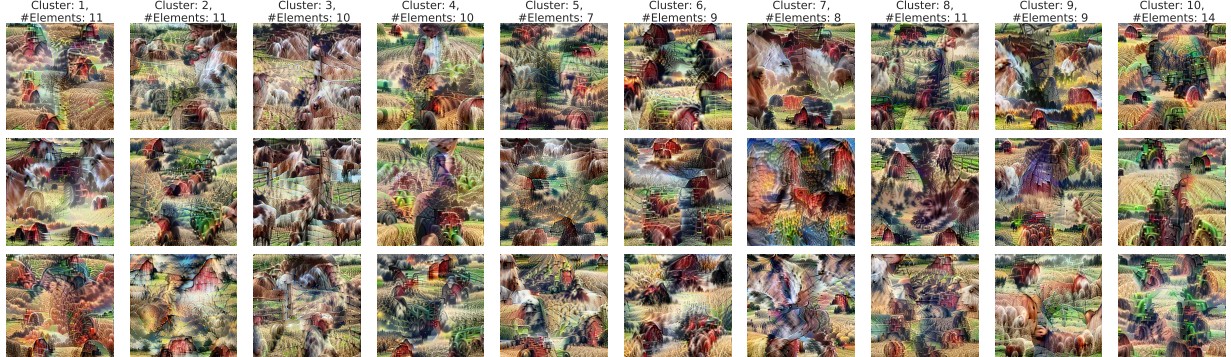

Figure 12: MultiSWAG clusters. The MultiSWAG clusters mostly cluster with regard to the underlying SWAG ensemble members. We can see, that most clusters contain a variety of *Farm* concepts in them.

## D  Multimodal structure of the posterior distribution of BNNs - WRes$0.7$ and WRes$2$.

In Figure 13 we show the tSNE plots and respective FVs of some example clusters of the WRes0.7 and WRes2 models. As can be seen, the FVs are naturally clustered together and well separated for the narrower WRes0.7 model, and overlap more for the WRes2 model.

## E  Additional experiment on the width dependence

Here we conducted the same experiment as in Section 4.5 with different backbone networks for defining the distance measure in the FV space, and on different datasets. The left panel of Figure 14 shows the dependence of the inter- (solid lines) and intra-mode (dashed lines) variances on the network width on CIFAR100, STL-10, and SVHN, when ResNet18, ResNet34 and ResNet50 are used as the backbone network. For the CIFAR-100 data set, we base our FV analysis on the class label *Castle*, for STL-10 on the class label *Bird*, and for the SVHN data set on the class label *8*. However, as the consistency across data sets for our results shows, any class label could potentially be used for this analysis. We can observe that, with all three backbone networks and on all three datasets, the inter-mode variance decreases with growing width (x-axis), while the intra-mode variance increases. These results are also reflected in the Feature Visualizations of these different-width models as shown in Figure 15, where we show the FVs for the *Bird* class of three different modes trained on the STL-10 dataset, and in Figure 16 where we show the FVs for the *8* class of three different modes trained on the SVHN dataset.

## F  Comparing pre-trained models to models that were trained from scratch.

In this section, we compare the FVs of models trained in two different scenarios. In the *pre-trained* scenario, pre-trained models are finetuned on the target dataset, while in the *from-scratch scenario*, training is performed from scratch. In the pre-trained scenario, we collected five ResNet50 models from Ashukha et al. (2020), which were pre-trained on ImageNet from different initializations, and finetuned them on CIFAR10 using the KFAC method. In the from-scratch scenario, we train five ResNet50 models from scratch on CIFAR10 using KFAC. As seen in Figure 17, FVs of class *Truck* for both scenarios look qualitatively similar. The left subplot of Figure 18 shows the posterior mode structure as a 2D projection onto the first two t-SNE components of the latent concept vectors obtained from each mode. Again, the cluster structure for both the *pre-trained* and *from-scratch* scenario looks similar. Moreover, the right subplot of Figure 18 shows FVVar of single mode, as well as all modes, for both training scenarios, which quantitatively confirms that the FV distribution is not significantly affected by the training scenario.

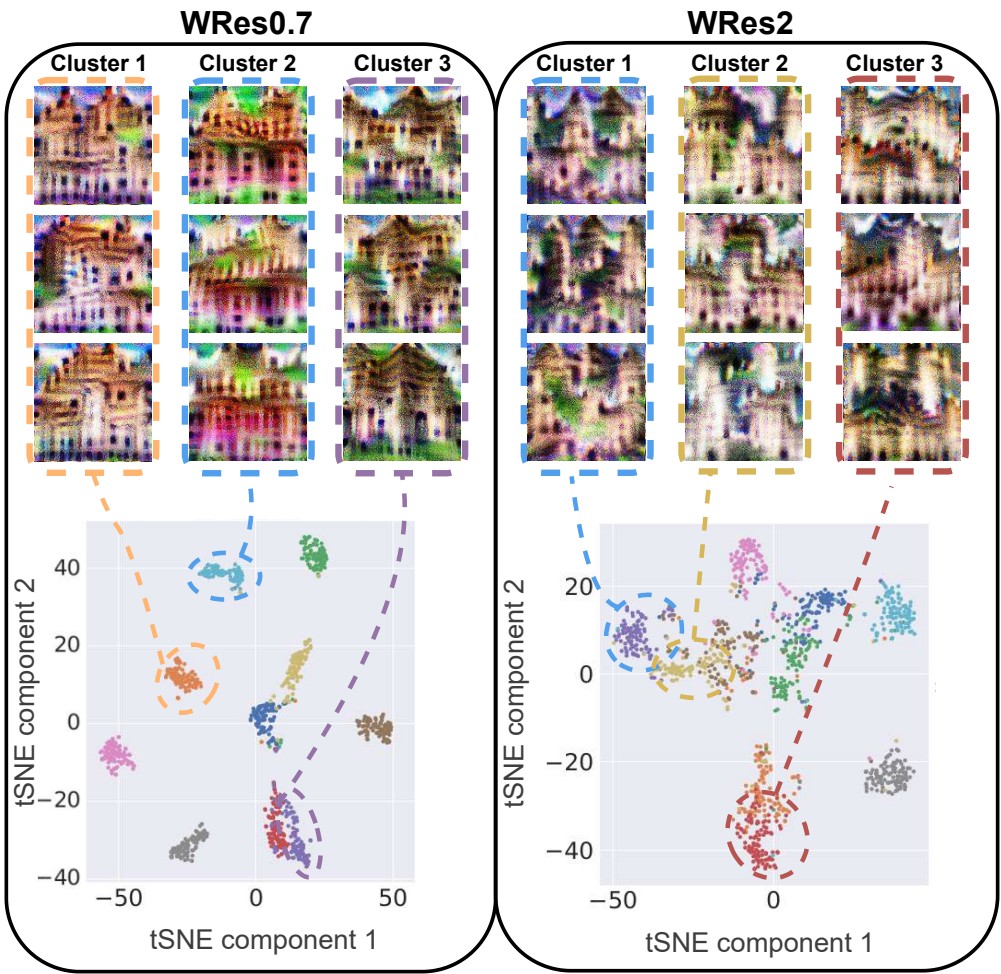

Figure 13: Multimodal structure of WReNet28 with different network widths in the FV space. The black bounding box corresponds to WRes0.7 and WRes2. The colored dashed bounding boxes mark FVs of 3 BNN instances from 3 modes. As the network width increases, the modes overlap more.

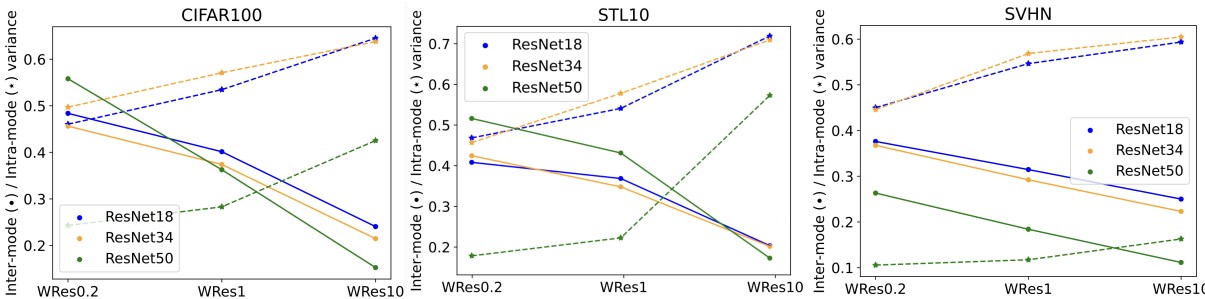

Figure 14: Inter- and intra mode variances on CIFAR100 (left), STL-10 (middle), and SVHN (right) using different ResNet backbones.

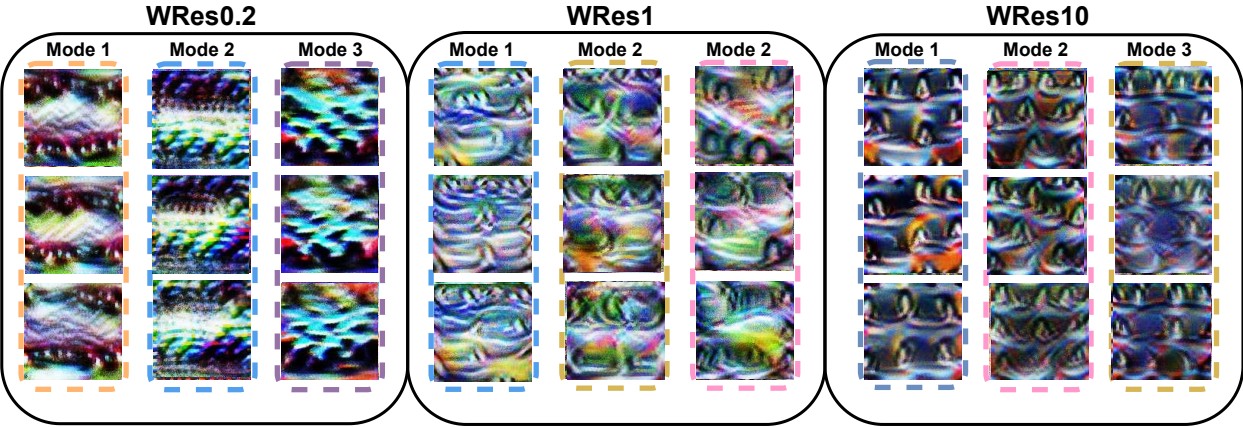

Figure 15: Feature Visualizations of class *Bird* that we sample from 3 different modes, e.g. MultiSWAG members, that were trained on the STL-10 dataset. We can see that the inter-mode variance decreases, and the intra-mode variance increases, when increasing the network width from left to right.

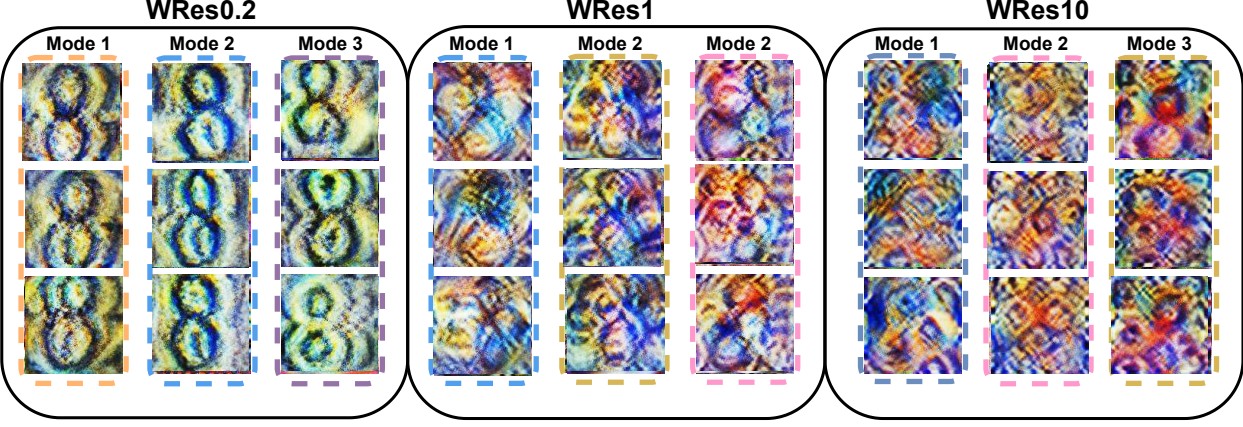

Figure 16: Feature Visualizations of class *8* that we sample from 3 different modes, e.g. MultiSWAG members, that were trained on the SVHN dataset. We can see that the inter-mode variance decreases, and the intra-mode variance increases, when increasing the network width from left to right.

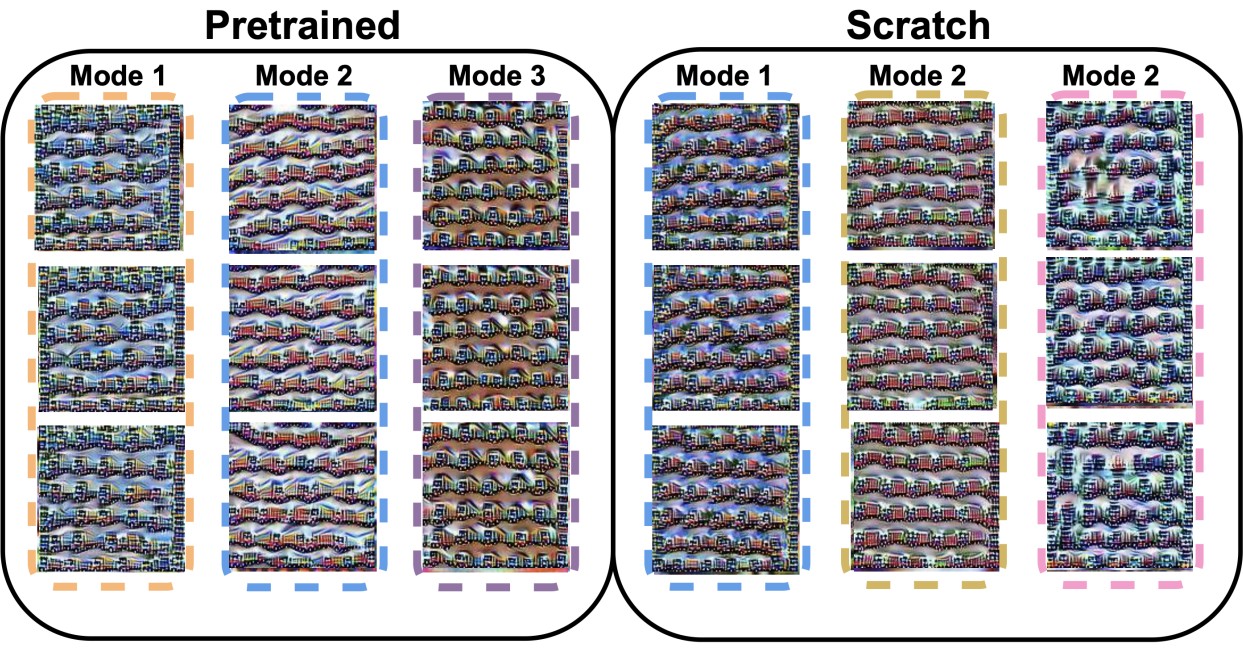

Figure 17: Comparison of Feature Visualizations from three different modes, e.g. KFAC models, that were first pre-trained on ImageNet and then fine-tuned on CIFAR10 (left), and three different modes that were trained from scratch on CIFAR10 (right). The concepts that are present in the FVs look similar for both pre-trained and from-scratch scenarios.

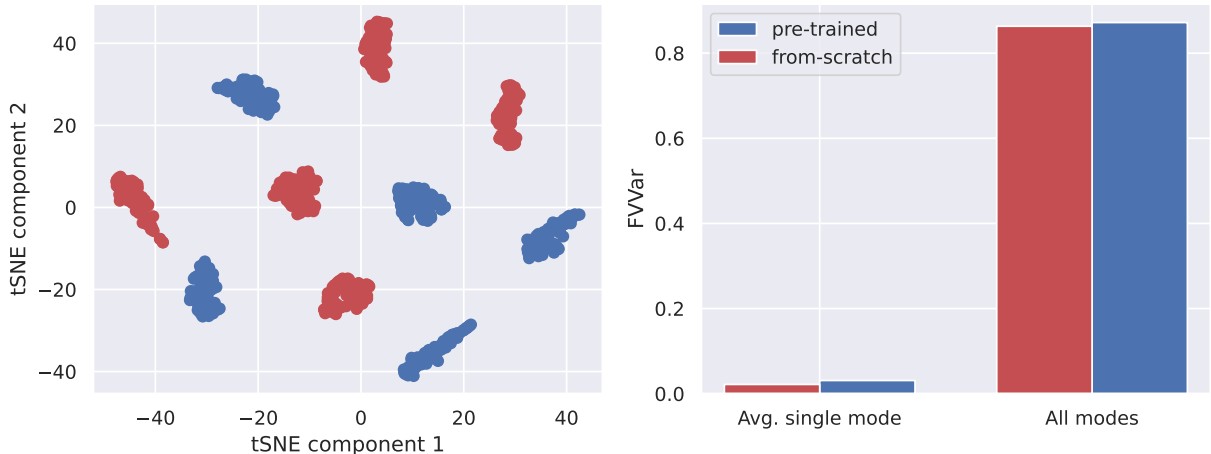

Figure 18: Multimodal structure (left) and FVVar (right) of KFAC models trained in the pre-trained (blue) and the from-scratch scenarios (red). The FVs of the models trained in both scenarios have qualitatively similar cluster structures, and quantitatively similar single-mode and all modes FVVar values.

