# OpenReview forum: "Visualizing the Diversity of Representations Learned by Bayesian Neural Networks"
_TMLR — Accepted by TMLR_

### Review · Reviewer_WYUH · 2023-04-05

**Summary Of Contributions:**

The paper looks at the question of explaining the diversity in the decision making strategies of Bayesian neural networks. While prior works looked at local explanations for BNNs ("Explaining bayesian neural networks", Bykov et al 2021), the authors focus here on global explanations. They conduct a study along these 4 axes:
 1. Whether the diversity of BNN samples can be explained with global explanations
 2. The impact of the Bayesian inference method on the explanation qualitatively and quantitatively
 3. The correlation between the diversity of explanations and the uncertainty estimates of BNNs
 4. The impact of architectural parameters on the explanations (the authors focus on the network width)

In order to conduct this investigation the authors chose the following setting:
* Architectures and datasets: WideResNet28 (with varying width) for CIFAR100 and ResNet50 for Place365. To vary the width, the authors multiply the number of channels of the WideResNet by 0.2, 0.7, 1, 2 and 10.
* Bayesian inference method: The authors conduct the study with different methods, either leading to a unimodal and a multimodal posterior: KFAC, Monte Carlo Dropout, Stochastic Weight Averaging Gaussian (SWAG) and its ensembled version (MultiSWAG).
* Explanation method: The authors chose to focus on  the activation maximization (AM) approach ("Synthesizing the preferred inputs for neurons in neural networks via deep generator networks", Nguyen et al. 2016) and regularize it with randomly transforming the input with rotation, scaling and jittering. This approach produce visualization of concepts (of the same size of the input images) that the network instance relies on to make a decision, called Feature Visualization (FV). For interpretability reasons, the FVs obtained from CIFAR100 networks are upsized to 128x128.

The authors also propose a distance measure for FVs in order to conduct quantitative analyses. The main challenge here is that naive distances could not reflect the invariance of the concepts to translation and rotation. In order to alleviate this, the authors learn to embed FVs into a lower dimensional space with a ResNet18 trained with contrastive loss (SimCLR framework). They compare FVs based on their cosine similarity in this latent space.

These settings and components are used in different experiments to show the validity of using FVs to explain the decision making of BNNs and reflect their properties reasonably, and to analyse the BNNs diversity and its dependence on Bayesian inference methods and certain architectural choices. For the first set of experiments, they qualitatively analyse FVs and show that they can use them to manipulate test samples in an interesting way, they study the impact of the AM algorithm hyperparameters on the visualizations and quantitatively correlate FVs with BNN uncertainty. In the second set of experiments, they show how the different modelling choices impact the diversity of the FVs and how they cluster in the latent space. This analysis corroborate recent observations and theories for DNNs. For example, they show a high correlation between the predictive entropy obtained with different Bayesian inference methods and the diversity of their FVs. They also provide a novel visualization of the impact of width on the network behavior, in line with recent findings based on Neural Tangent Kernel analysis.

**Audience:**

Yes

**Broader Impact Concerns:**

I don't think there are any concerns from the ethical implications side in this work. However, I would encourage the authors to consider how their contribution can be used to analyse other aspects beyond the network uncertainty, such us bias.

**Claims And Evidence:**

No

**Requested Changes:**

Critical changes:
As mentioned above, the major change that I think is needed to validate the findings of this paper is to remove the gap between the analysed networks and the embedding networks complexities. There two somehow obvious choices for this:
* Either use the same architecture both for the BNNs and for the distance measure,
* Or bring back the distance to the pixel space. The invariance issue can be dealt with by a proper use of data augmentations.

Strengthening suggestions:
* What happens if we vary the depth of the network?
* Does this analysis generalize well to other families of networks (e.g. transformers)?
* What is the impact of the training regime on this analysis? More precisely, my understanding is that the networks used here are trained from scratch on the selected datasets. What happens when the training starts from a pretrained network instead?

**Strengths And Weaknesses:**

Strengths:
* The paper is clearly written and well structured (modulo a few needed corrections listed below)
* The general approach is reasonable, and the experimentation is well designed and leads to interesting insights:
     * I particularly appreciated the first experiment where the authors showed that the network instances that wrongly classify a test image are focusing on a concept that is not present in the image, and how manipulating the test image by adding a small component from this misleading concept leads to a correct prediction.
     * I also appreciated the analysis of the impact of width and the Bayesian inference methods on the FVs and linking it to the different networks behavior and to recent DNN theory.
* The authors provide a plethora of details (including the public repositories they used in their study), making the work highly unpredictable.

Weakness:
* Minor corrections:
   * Better separate Figure 1 caption and the main text.
   * Paragraph 4.5: There is a lot of redundancy in this paragraph.
   * Eq (7): The authors gives the definition of cosine similarity instead of cosine distance. Can they confirm that the correct formula has been used in their experiments?
* The major weakness in my opinion lies in the choice the authors made to measure the distance in the FV space. While the rationale of the choice makes a lot of sense, the distance measure is highly depending on the expressivity of ResNet18, and is then applies to inputs of variable complexity. This can make the analysis misleading, e.g. the higher inter-mode variance and lower intra-mode variance observed for wider models in Figure 7 can be a result of the incapabilities of the ResNet to disentangle them more than the properties of the FVs. Indeed, we can visually see that the FVs of the wider networks are of higher definition. See next section for suggestions.
* The authors focus on the analysis of width on the FV diversity, and the choice of the Bayesian inference method. There are a plenty of other phenomenons that could be interesting to study for a more complete and impactful contribution. See next section for suggestion.

---

> ### Author Response · Authors · 2023-08-09
> **Answer to reviewer "WYUH"**
>
> We would like to express our sincere appreciation to Reviewer “WYUH” for constructive and insightful comments. We are pleased to hear that the reviewer found our approach and experimentation clearly explained and reasonable, as well as our insights interesting.
>
> Below we give our reply to each comment:
>
> **Comment:**
> 1.1: The major weakness in my opinion lies in the choice the authors made to measure the distance in the FV space. While the rationale of the choice makes a lot of sense, the distance measure is highly depending on the expressivity of ResNet18, and is then applies to inputs of variable complexity. This can make the analysis misleading, e.g. the higher inter-mode variance and lower intra-mode variance observed for wider models in Figure 7 can be a result of the incapabilities of the ResNet to disentangle them more than the properties of the FVs. Indeed, we can visually see that the FVs of the wider networks are of higher definition. See next section for suggestions.
> Critical changes: As mentioned above, the major change that I think is needed to validate the findings of this paper is to remove the gap between the analysed networks and the embedding networks complexities. There two somehow obvious choices for this:
> Either use the same architecture both for the BNNs and for the distance measure,
> Or bring back the distance to the pixel space. The invariance issue can be dealt with by a proper use of data augmentations.
>
> **Reply:**
> We used ResNet18 as a backbone network for our distance metric because it was shown in [1] that the feature extractor of a ResNet18 is sufficiently powerful to extract discriminative features for ImageNet10 and CIFAR10 datasets. However, to address the reviewer’s concern, we tested other backbone networks.  Specifically, in Appendix D in the revision, we conducted the same experiment with ResNet18, ResNet34, and ResNet50 as the backbone network on CIFAR100, as well as STL10[2] and SVHN[3], and observed similar tendency, i.e., when increasing the network width, the inter-mode variability decreases, while the intra-mode variability increases.  This results imply that the observed tendency, which aligns the deep learning theory, is not because of the limited expressivity of ResNet18.
>
> ***References:***
>
> [1] Li, Yunfan & Hu, Peng & Liu, Zitao & Peng, Dezhong & Zhou, Joey & Peng, Xi. (2021). Contrastive Clustering. Proceedings of the AAAI Conference on Artificial Intelligence. 35. 8547-8555. 10.1609/aaai.v35i10.17037
>
> [2] Coates, Adam & Lee, Honglak & Ng, Andrew. (2011). An Analysis of Single-Layer Networks in Unsupervised Feature Learning. 1-9.
>
> [3] Netzer, Yuval & Wang, Tao & Coates, Adam & Bissacco, Alessandro & Wu, Bo & Ng, Andrew. (2011). Reading Digits in Natural Images with Unsupervised Feature Learning.
>
>
> **Comment:** The authors focus on the analysis of width on the FV diversity, and the choice of the Bayesian inference method. There are a plenty of other phenomenons that could be interesting to study for a more complete and impactful contribution. See next section for suggestion. What happens if we vary the depth of the network? Does this analysis generalize well to other families of networks (e.g. transformers)? What is the impact of the training regime on this analysis? More precisely, my understanding is that the networks used here are trained from scratch on the selected datasets. What happens when the training starts from a pre-trained network instead?
>
> **Reply:** Thank you for the interesting suggestions.  In this paper, we propose a novel analysis tool, and demonstrated its usefulness by answering to the simplest questions, including the choice of inference methods, and the network width dependence.  Unlike analyzing the width, where we can simply multiply the layer width by some factor, analyzing depth requires more careful consideration especially when the network has some special architectures, e.g., skip-connections in ResNets. Therefore, we will leave those extended ideas of analysis as future work.  We plan to analyze different network architectures, e.g., Transformers, in the near future.
>
> **Comment:** Better separate the Figure 1 caption and the main text.
>
> **Reply:** Fixed.
>
> **Comment:** Paragraph 4.5: There is a lot of redundancy in this paragraph.
>
> **Reply:** We have revised section 4.5 and have removed redundancies particularly when explaining the concepts between the FVs of different modes of different width networks, and how these relate to the ensemble test accuracies and test ECEs.
>
> **Comment:** Eq (7): The authors give the definition of cosine similarity instead of cosine distance. Can they confirm that the correct formula has been used in their experiments?
>
> **Reply:** We confirm that we used the cosine-similarity, i.e., Eq. (7), in all experiments and have fixed the typo.

---

> > ### Comment · Reviewer_gAeF · 2023-08-16
> > **Question about response to WYUH**
> >
> > Dear authors,
> >
> > While reading your response to WYUH I noticed you left these two questions unanswered:
> >
> > > What is the impact of the training regime on this analysis? More precisely, my understanding is that the networks used here are trained from scratch on the selected datasets. What happens when the training starts from a pre-trained network instead?
> >
> > I am actually curious about these two points too, would you mind providing more details?
> >
> > Thanks!

---

> > > ### Author Response · Authors · 2023-08-23
> > > **Answer to question of gAeF**
> > >
> > > Dear reviewer “gAeF”,
> > >
> > > Thank you for your comment. We could conduct an experiment comparing a
> > > trained model from random initialization vs. a fine-tuned model on pre-trained weights for KFAC. Specifically, we could use any modern model CNN architecture, pre-trained on ImageNet and finetune it on CIFAR10, for example. Could you please tell us if this is what you are curious of? For the other Bayesian learning methods, i.e., MCDO, SWAG, and MultiSWAG, there is no out-of-the-box pre-trained model, to the best of our knowledge.

---

> > > > ### Comment · Reviewer_gAeF · 2023-09-08
> > > > **Answer to authors**
> > > >
> > > > > We could conduct an experiment comparing a trained model from random initialization vs. a fine-tuned model on pre-trained weights for KFAC. Specifically, we could use any modern model CNN architecture, pre-trained on ImageNet and finetune it on CIFAR10, for example.
> > > >
> > > > That would be very interesting! And I believe it would answer WYUH's concern (although best would be that WYUH confirmed that).

---

### Review · Reviewer_gAeF · 2023-04-21

**Summary Of Contributions:**

The authors analyze feature visualization (FV) techniques on Bayesian Neural Networks (BNNs). They show that 1) FV can be used on BNNs, 2) With the help of contrastive learning to define a distance measure between FVs, they show that different BNN methods yield different levels of diversity in their FV, 3) the diversity of FVs correlates with uncertainty measures, and 4) wider networks tend to reduce FV diversity between different weight modes while it increases within a mode. The authors complement these points with illustrative qualitative results on the produced FVs, like Figure 1.

**Audience:**

Yes

**Broader Impact Concerns:**

I do not foresee any relevant ethical concern from this work.

**Claims And Evidence:**

Yes

**Requested Changes:**

* Clarify how the FV dataset is built to train SimCLR
* Fix space after Figure 1
* Explain the choice of datasets
* Fix other minor problems (see above)

**Strengths And Weaknesses:**

Overall Review
===========
I found this work very interesting, particularly how neural network width interacts with multi-modal Bayesian solutions and how it relates to the NTK. In that sense, I think this work would be of interest to the research community and it should be accepted.

Strengths
=======
* XAI is an increasingly important topic as ML becomes widespread in real-life applications.
* The comparison between different Bayesian estimator is interesting
* The paper is well-written and easy to read

Weaknesses
=========
* The way the contrastive learning method is trained is not clear to me. If I understand it well, you train a resnet18 with SimCLR on a dataset of FVs. Do you train one resnet per Bayesian network? How do you do it for MultiSWAG?
* Figure 1's caption is too close to the text below, which makes readers lose time looking for where the text continues.
* Could you give some details on why the choice of Places and CIFAR?

**Minor**
* There are many mentions to *Place365*. It should be *Places365* instead.
* Section 4.1: *which is classify correctly* -> *which is classified correctly*

---

> ### Author Response · Authors · 2023-08-09
> **Answer to reviewer "gAeF"**
>
> We express our sincere appreciation to Reviewer "gAeF" for valuable comments. We are delighted to learn that the reviewer deems our work important, engaging, and well-written.
>
> Below we give our reply to each comment:
>
> **Comment:** The way the contrastive learning method is trained is not clear to me. If I understand it well, you train a resnet18 with SimCLR on a dataset of FVs. Do you train one resnet per Bayesian network? How do you do it for MultiSWAG? Clarify how the FV dataset is built to train SimCLR.
>
> **Reply:** For our distance measure, we train one contrastive learning model on all FVs generated by the respective Bayesian inference method. For MultiSWAG, we train one contrastive learning model on the collection of FVs that are generated by all ensemble members, e.g. modes. In particular, we generate and train on 1000 FVs for each Bayesian inference method, including MultiSWAG, where we generate 100 FVs per member (10 total members), and thus also train on a total of 1000 FVs. We have clarified this in Section 3.5 in our revised paper.
>
> **Comment:** Explain the choice of datasets. Could you give some details on why the choice of Places and CIFAR?
>
> **Reply:** We chose Places365 and CIFAR100 for the following reasons: Places365 is preferred because it encompasses numerous classes with diverse human-understandable concepts, such as the farm class, which typically includes concepts like farm animals, barns, fields, and others. CIFAR100 was chosen in our Mixture-of-Gaussian experiments due to its size and variety of concepts. In Appendix D in the revision, we also conducted the experiments on inter-/ intra-mode variability in Section 4.5, for two more datasets, STL10[1] and SVHN[2], which further supports our statements.
>
> ***References:***
>
> [1] Coates, Adam & Lee, Honglak & Ng, Andrew. (2011). An Analysis of Single-Layer Networks in Unsupervised Feature Learning. 1-9.
>
> [2] Netzer, Yuval & Wang, Tao & Coates, Adam & Bissacco, Alessandro & Wu, Bo & Ng, Andrew. (2011). Reading Digits in Natural Images with Unsupervised Feature Learning.
>
> **Comment:** Fix space after Figure 1.
>
> **Reply:** Done.
>
> **Comment:** Section 4.1: which is classify correctly -> which is classified correctly
>
> **Reply:** Fixed.
>
> **Comment:** There are many mentions to Place365. It should be Places365 instead.
>
> **Reply:** Fixed.

---

### Review · Reviewer_FCqy · 2023-07-10

**Summary Of Contributions:**

The paper aims at improving understanding of the decision-making
strategies of BNNs in the realm of Explainable AI. It provides qualitative insights through feature
visualizations and quantitaive insights by contrastively learned
distance metric.

Feature visualizations are provided through the use of the Activation Maximization
framework. These features need to be mapped to latent representations in
order to be quantitatively anlyzed. This mapping is learned through SimCLR.

The paper gives answers to the following questions:
1. Can the diversity of BNN instances be explained by global XAI methods?
2. Does the choice of the Bayesian inference method affects the diversity of their feature visualization?
3. Can the uncertainty estimates provided by a BNN be explained by the diversity of their feature
visualizations?
4. How does the network width affects the diversity of explanations of samples from a multimodal
posterior distribution?

**Audience:**

Yes

**Broader Impact Concerns:**

None.

**Claims And Evidence:**

No

**Requested Changes:**

The underlying motivation for the relevant directions the paper investigates needs to be made clearer (or maybe only tackle two out of the 4 questions mentioned in the introduction). For a strong experiment paper, much more datasets need to investigated with the presented methodology, much more concepts (in the way the paper uses it) need to be presented.

**Strengths And Weaknesses:**

Strength
* The paper tackles a generally important problem with interesting insights: How can the behaviour of BNNs understood such that it's interpretable/exaplainable.
* A strong empirical result is their demonstration that different Bayesian inference methods affects the diversity of BNN instances.

Weakness
* The four questions the authors pose in the introduction seem only weakly related. Why would one tackle those 4 aspects jointly? Specifically, what does a very specific architecture related aspect like the width of a network have to do with the other rather generic questions.
* The paper presents itself as a mostly qualitative type of work. For this however, the breadth of experiments/datasets seems to be too small. The experiments shown in the paper could be easily cherrypicked.

---

> ### Author Response · Authors · 2023-08-09
> **Answer to reviewer " FCqy"**
>
> We sincerely thank Reviewer “FCqy” for the insightful comments, which allowed us to refine our paper and present our research directions more explicitly.  We are delighted to receive positive feedback on our endeavor to elucidate the workings of Bayesian Neural Networks (BNNs) in a manner understandable to humans. Additionally, we are grateful for your acknowledgment of the significance of our experiments, where we explore how different Bayesian inference methods influence the diversity of global explanations in BNN instances.
>
> Below we give our reply to each comment:
>
> **Comment:** The four questions the authors pose in the introduction seem only weakly related. Why would one tackle those 4 aspects jointly? Specifically, what does a very specific architecture related aspect like the width of a network have to do with the other rather generic questions. The underlying motivation for the relevant directions the paper investigates needs to be made clearer (or maybe only tackle two out of the 4 questions mentioned in the introduction).
>
> **Reply:** Our paper introduces a method for analyzing different facets of BNNs. To address this objective, we selected four specific research directions, and the rationale behind these choices is elaborated upon below.
> The first question is about the basic requirement.  Our approach cannot answer any question unless the empirical diversity of BNN samples is reflected in their explanations in a human-understandable way.
> The other three questions are related to the current hot topics in Bayesian deep learning, i.e., posterior approximation, uncertainty estimation, and the loss surface of neural networks.  Since the Bayesian posterior inference is hard for deep neural networks, researchers have been proposing many methods, which show different properties.  The most important property that Bayesian learning offers is uncertainty estimation, for which no existing methods provide satisfactory performance.  The state-of-the-art Bayesian learning method for deep neural networks is deep ensemble and its extensions, including the methods finding continuous solution regions on which the training losses are small.  We approach to those challenges by visualizing the learned representations, i.e., feature visualizations.  Our experiments showed that the diversity of different posterior approximation methods reflects the diversity of the representations, which is well correlated to the uncertainty estimation.  Our investigation of the inter-/intra-mode diversities of mixture-of-Bayesian neural networks showed that the theoretically proven phenomenon—infinitely wide network has equivalent solutions close to any initial points [1]---can be observed through the lens of feature representations. We added this motivation in the introduction, and believe that researchers in Bayesian deep learning and explanability are interested in our results.
>
> **Comment:**
> The paper presents itself as a mostly qualitative type of work. For this, however, the breadth of experiments/datasets seems to be too small. The experiments shown in the paper could be easily cherry-picked. For a strong experiment paper, much more datasets need to be investigated with the presented methodology, much more concepts (in the way the paper uses it) need to be presented.
>
> **Reply:**
> We have tested our approach on 2 more datasets, STL10[2] and SVHN[3], and two more backbone networks for out distance measure, e.g. ResNet34 and ResNet50, and present a comparison for all three datasets (CIFAR100 included) in Appendix D.  Irrespective of the vision domain dataset and the backbone network for the feature visualization distance measure, we observe the same trend, i.e., when the network width increases, the inter-mode variability decreases, while the intra-mode variability increases.
> The other questions we addressed in the paper were supported by quantitative experiments based on the entire test dataset for uncertainty experiments (question 3) and several hundred Feature Visualizations for Bayesian inference (question 2).
>
> ***References:***
>
> [1] Jacot, Arthur & Gabriel, Franck & Hongler, Clément. (2018). Neural Tangent Kernel: Convergence and Generalization in Neural Networks.
>
> [2] Coates, Adam & Lee, Honglak & Ng, Andrew. (2011). An Analysis of Single-Layer Networks in Unsupervised Feature Learning. 1-9.
>
> [3] Netzer, Yuval & Wang, Tao & Coates, Adam & Bissacco, Alessandro & Wu, Bo & Ng, Andrew. (2011). Reading Digits in Natural Images with Unsupervised Feature Learning.

---

### Author Response · Authors · 2023-07-25
**Official comment to all reviewers**

We express our sincere appreciation to the reviewers for their thorough evaluation of our manuscript. We believe that their invaluable suggestions provided will profoundly augment the quality of our research. Unfortunately, due to the maintenance of our computational infrastructure we could not have finished our additional experiments by the due date. Therefore, we kindly ask for an extension of the deadline for revision and reply comments to Sunday, August 6th of 2023. Once the revised paper is uploaded we will notify all reviewers, and comment each of their reviews accordingly.

---

> ### Comment · Action_Editors · 2023-07-25
> **Re: Official comment to all reviewers**
>
> Dear authors, that is not a problem. Thank you for informing us.  Looking forward to the replies and revisions.

---

### Author Response · Authors · 2023-08-09
**Official comment to all reviewers**

We thank the reviewers for their constructive comments, which have profoundly augmented the quality of our research.
We were delighted to hear that all reviewers found our work to be well motivated and important, that our paper was clearly explained and reasonable (“WYUH” and "gAeF"), and that our insights are of great interest (“WYUH”,  “gAeF”, “FCqy”).

We have uploaded a revision and provided detailed answers to the comments as official comments to each reviewer. The major changes of the revisions are:

- We have added a comparison of inter- and intra-mode variability of different backbone network architectures for our distance measure in Appendix D.
- We have added a comparison of Feature Visualizations, inter- and intra-mode variability of different datasets for our different width experiments in Appendix D.
- We have fixed the spacing of the Figure 1 caption.
- We have revised paragraph 4.5 and removed redundancies.
- We have confirmed the right choice of the cosine-similarity for all of our experiments and have fixed the corresponding typos.
- We have provided the detailed procedures of our contrastive learning models in section 3.5.
- We have added an additional explanation for our dataset choices in section 3.1.
- We have fixed the ‘classified’ typo in section 4.1.
- We have fixed the Places365 dataset typo.
- We added motivations to the four questions in the introduction.
- We have added two more datasets and two more contrastive model backbone networks for our width-experiments in Appendix D.

We believe that our replies address all concerns raised by the reviewers, and the revision satisfies the two TMLR criteria.

---

### Decision · Action_Editors · 2023-09-29

**Recommendation:** Accept with minor revision

**Comment:**

While the breadth of experiments was modest at first, the authors conducted a large number of additional experiments based on the reviewer's input. Reviewers were satisfied with the revision and additional material provided by the authors. They also found the insights valuable and worth publishing.

I recommend acceptance provided the authors conduct the additional experiment discussed with reviewer gAeF.

**Audience:**

The work is of interest to the TMLR community. XAI is an important topic and its applications to BNNs is less common.

**Claims And Evidence:**

This paper offers a qualitative study of the applications of explainable AI methods to learned representations in Bayesian neural networks.
The work is clearly motivated and the methodology sound. While the novelty is modest, the insights from this study are worth publishing as   indicated by the reviewers.

---

> ### Author Response · Authors · 2023-11-01
> **Answer to the decision of the action editors**
>
> Dear action editors and reviewers,
>
> We would like to thank the action editors for their decision and are pleased to announce that we have included the requested experimental results for the impact of pre-training in Appendix F of our camera-ready version. We refer to these results in the second paragraph of Section 4.5 in the manuscript.